# EIT: Enhanced Interactive Transformer

## Abstract

Two principles: the *complementary principle* and the *consensus principle* are widely acknowledged in the literature of multi-view learning. However, the current design of Multi-head self-attention, an instance of multi-view learning, prioritizes the complementarity while ignoring the consensus. To address this problem, we propose an enhanced multi-head self-attention (EMHA). First, to satisfy the *complementary principle*, EMHA removes the one-to-one mapping constraint among queries and keys in multiple subspaces and allows each query to attend to multiple keys. On top of that, we develop a method to fully encourage consensus among heads by introducing two interaction models, namely Inner-Subspace Interaction and Cross-Subspace Interaction. Extensive experiments on a wide range of language tasks (e.g., machine translation, abstractive summarization and grammar correction, language modeling), show its superiority, with a very modest increase in model size.

## 1 Introduction

Transformer architectures (Vaswani et al., 2017) have yielded promising results on a wide range of natural language processing tasks (Devlin et al., 2019; Brown et al., 2020). A key factor contributing to their success is the multi-head self-attention network (MHSA), which enables efficient modeling of global dependencies among tokens in parallel. Notably, instead of utilizing a single attention mechanism, MHSA uses an ensemble of attention models, each models a small subspace, and finally aggregates these results to the final one. The core idea is similar to subspace learning (Blum and Mitchell, 1998) or multi-view learning (Chaudhuri et al., 2009).

In the realm of multi-view learning, two fundamental principles guide the research: the *complementary principle* and the *consensus principle* (Xu et al., 2013). The *complementary principle* emphasizes that each data view may possess unique knowledge not present in other views, prompting the use of multiple views for a comprehensive and accurate data description. On the contrary, the *consensus principle* aims to maximize the agreement on multiple distinct views. However, in the context of MHSA design, most studies predominantly focus on the *complementary principle*. This oversight is evident in their encouragement of diverse information capture by different heads (Li et al., 2018; Cui et al., 2019) and the adoption of complex aggregation operations (Li et al., 2019; Wang and Tu, 2020). Some studies (Michel et al., 2019; Clark et al., 2019; Voita et al., 2019; Behnke and Heafield, 2020) even consider the high similarity among attention heads as a significant problem referred to as *attention redundancy*.

Although diversity is crucial in multi-view learning, Dasgupta et al. (2001) has shown that simply fusing diverse outputs does not guarantee improved results: the probability of a disagreement of two independent hypotheses upper bounds the error rate of either hypothesis. The *consensus principle* highlights the need to minimize disagreement among views to achieve better outcomes. In response to the *consensus principle*, several studies (Kumar and III, 2011; Kumar et al., 2011) have focused on minimizing disagreement among views to achieve better outcomes. However, in the context of MHSA research, there is a tendency to prioritize complementarity over consensus among different attention heads. Here we ask a question: *Can striking a balance between these two principles be beneficial for designing MHSA mechanisms?*

However, encouraging such a consensus in multi-head self-attention is challenging. In our preliminary experiments, we find that directly utilizing regularization terms can achieve this goal but cannot improve performance. Drawing inspirations from human behavior where group discussions and interactions foster consensus, we propose intro-

ducing interactions among different subspaces in MHSA to achieve consensus.

To this end, we propose a new multi-head self-attention variant: Enhanced Multi-Head Self-Attention, which encourages the consensus among attention heads while guaranteeing to contain sufficient information. To ensure information sufficiency, we propose a novel many-to-many mapping scheme to generate numerous high-quality initial attention maps. This can generate more attention maps without suffering low-bottleneck problems (Bhojanapalli et al., 2020). On top of these sufficient attention maps, we propose two interaction components: *inner-subspace interaction* (ISI) and *cross-subspace interaction* (CSI). These hierarchical interaction modules fully encourage consensus among attention maps of different heads.

The outcome of this work is an Enhanced Interactive Transformer (EIT) architecture in that MHSA is replaced with Enhanced Multi-Head Attention (EMHA). Our proposed EIT has been demonstrated to be simple to implement and highly parameter efficient, yet it consistently produces impressive results across a diverse set of tasks, including machine translation, grammar error correction, abstractive summarization, and language modeling. In addition, we have developed a computationally efficient variant of EIT, which, while still maintaining strong performance on several tasks, is better suited for low-latency industrial applications.

## 2 Preliminary: Multi-Head Self-Attention

Multi-head self-attention (MHSA) is an efficient operation that can capture the interactions among tokens. Given an embedded input sequence $\mathbf{X} \in \mathbb{R}^{T \times d}$, MHSA is defined as follows:

$$\mathbf{A}^i = \text{Softmax}(\frac{(\mathbf{X}\mathbf{W}_Q^i)(\mathbf{X}\mathbf{W}_K^i)^\mathsf{T}}{d^k}) \quad (1)$$

$$\mathbf{O} = \sum_{i=1}^{M} \mathbf{A}^i \mathbf{X}\mathbf{W}_V^i \mathbf{W}_O^i \quad (2)$$

where $T$ denotes the sequence length, $d$ is the input embedding dimension, $d_k$ is the head dimension, $M$ is the number of head partition on representations, $\mathbf{W}_Q^i, \mathbf{W}_K^i, \mathbf{W}_O^i \in \mathbb{R}^{d \times d_k}$, $\mathbf{W}_O^i \in \mathbb{R}^{d_k \times d}$. $\mathbf{A}^i$ represents the attention distribution of $i$-th head. Without special declaration, we use $\mathbf{Q}^i, \mathbf{K}^i, \mathbf{V}^i$ to refer to $\mathbf{X}\mathbf{W}_Q^i, \mathbf{X}\mathbf{W}_K^i, \mathbf{X}\mathbf{W}_V^i$, respectively, which denotes the query, key and value in $i$-th head.

## 3 Enhancing Consensus in Transformers

### 3.1 Enhanced Interactive Transformer

We design a novel Enhanced Interactive Transformer (EIT) in which we replace the multi-head self-attention with Enhanced Multi-Head Attention mechanism (EMHA) that encourages consensus among different attention heads.

### 3.1.1 Many-to-Many Mapping Scheme

Intuitively, to achieve better consensus, multi-head self-attention should first contain as much information as possible. To achieve this goal, a natural idea is to employ more attention heads in multi-head self-attention. However, multi-head self-attention with too many heads suffers from low bottleneck problem (Bhojanapalli et al., 2020), resulting in performance deterioration in practical applications.

Although various strategies like *attention expansion* (Shazeer et al., 2020; Zhou et al., 2021b) have been proposed, the information captured in their attention maps remains limited due to an additional linear transformation step, which can introduce similarity among the maps.

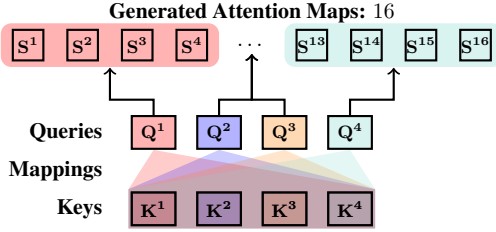

Figure 1: The illustration of many-to-many mapping scheme ($M = 4$).

To alleviate this problem, we propose a novel many-to-many (M2M) mapping scheme that enables each query to attend to $M$ keys instead of a single key. As illustrated in Figure 1, four queries and four keys can be served as two components in a bipartite graph and each element in a component, e.g., $\mathbf{Q}^1$, can interact with any elements in another component, e.g., $\mathbf{K}^1, \ldots, \mathbf{K}^4$. Formally, supposing one with $M$ heads, the $i$-th attention map can be formally calculated as:

$$\mathbf{S}^i = \frac{\mathbf{Q}^{\lfloor(i-1)/M+1\rfloor}(\mathbf{K}^{(i-1)\%M+1})^\mathsf{T}}{\sqrt{d_k}} \quad (3)$$

where $i \in \{1, \ldots, M^2\}$, $\mathbf{S}^i \in \mathbb{R}^{T \times T}$ is the attention maps without softmax, $\lfloor \rfloor$ is the round down operation and % is the mod operation. For example, $\mathbf{S}^4$ is computed by $\mathbf{Q}^1$ and $\mathbf{K}^4$ when $M$ equals to 4.

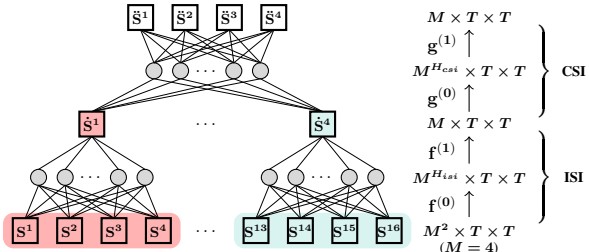

Figure 2: Illustration of dual enhanced interaction in EIT ($M = 4$). We omit the ReLU for simplicity.

**Discussion.** M2M demonstrates an increased capacity to generate M times the number of attention maps when given identical input. This enhanced capability can be attributed to effective utilization of a many-to-many mapping strategy by M2M, which fully leverages the original head features, such as $\mathbf{Q}$ and $\mathbf{K}$. Notably, this approach successfully avoids the production of similar attention maps by employing a dot-multiplication strategy to directly generate the attention maps (See Figure 6). By avoiding the generation of redundant attention maps, M2M improves its ability to capture diverse and distinct patterns in the input data. As a result, it facilitates the subsequent creation of more comprehensive and informative representations. This component can also be viewed as a strategy to enhance *complementary principle*.

### 3.1.2 Dual Enhanced Interaction

As aforementioned, M2M enlarges the information capacity, which provides a prerequisite for encouraging consensus among different heads. To encourage consensus, a simple idea is to directly add a linear transformation among attention maps (Shazeer et al., 2020; Zhou et al., 2021b; Wang and Tu, 2020). While these methods can achieve performance improvements in vanilla Transformer settings, they are unsuitable in our framework. One key factor is that our framework encompasses a wealth of information; however, it also incorporates certain elements of noise. Such a coarse level of interaction fails to attain a satisfactory consensus.

To address this problem, we propose a finer solution that is able to differentiate between relevant and irrelevant information, discarding the latter while fully capitalizing on the former. Two kinds of interactions among those attention maps are introduced hierarchically, the inner-subspace interaction and cross-subspace interaction.

**Two Relationships.** We begin with identifying two important relationships: inner-subspace interaction (ISI) relationship and cross-subspace interaction (CSI) relationship. As illustrated in Figure 1, the inner-subspace interaction (ISI) relationship describes the connection among the attention maps generated by the same query, e.g., attention maps in the block of same color. These attention maps own a closer relationship. The cross-subspace interaction (CSI) relationship describes the collaboration among different heads, which exists in the attention maps generated by different queries, e.g., attention maps from blocks of different color.

**Inner-Subspace Interaction Modeling.** One can adopt the standard convolution operation via batch transformation. However, such a way ignores the difference among the ISI relationship constrained by different queries, e.g., the ISI relationship in red block and blue block in Figure 1. It is desirable to preserve and enhance this distinction. To more efficiently model the interaction within subspaces, we therefore adopt group convolutions (Krizhevsky et al., 2012), which use separate parameters to process features from different groups.

Denote $\mathbf{f}(\cdot)$ as a single layer group convolution. As illustrated in Figure 2, given the output of M2M, namely $\mathbf{S}$, as input, ISI sub-module is computed as:

$$\dot{\mathbf{S}} = \mathbf{f^{(1)}}(\mathbf{ReLU}(\mathbf{f^{(0)}}(\mathbf{S}))) \tag{4}$$

where $\dot{\mathbf{S}} \in \mathbb{R}^{M \times T \times T}$ is the output of the ISI sub-module. We use $M^{H_{isi}}$ to represent the intermediate head size in ISI sub-module and set the number of groups in group convolutions to $M$.

Finally, we can obtain $M$ high-quality attention maps that effectively retain the benefits of using a larger number of attention heads while discarding irrelevant information. Such a process is another key for Transformer to benefit from more heads and is unique to our work.

**Cross-Subspace Interaction Modeling.** To efficiently model the cross-subspace interaction, we adopt two-layer convolutions accompanied by the ReLU activation to consist this sub-module.

Let us denote $\mathbf{g}(\cdot)$ as a single layer convolution. As illustrated in Figure 2, given the output of ISI, namely $\dot{\mathbf{S}}$, as input, CSI sub-module is computed as:

$$\ddot{\mathbf{S}} = \mathbf{g^{(1)}}(\mathbf{ReLU}(\mathbf{g^{(0)}}(\dot{\mathbf{S}}))) \tag{5}$$

where $\ddot{\mathbf{S}} \in \mathbb{R}^{M \times T \times T}$ is the output of the CSI sub-module. We use $M^{H_{csi}}$ to represent the intermedi-

ate head size in CSI sub-module. Finally, we can obtain $M$ final attention maps that fully leverage the benefits of each head.

## 3.2 Efficient Version of EIT

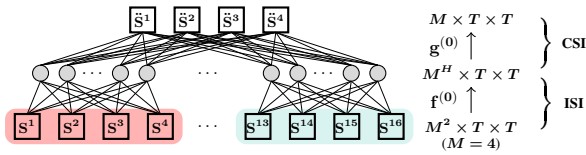

Figure 3: Illustration of dual enhanced interaction in efficient EIT ($M = 4$). We omit the ReLU for simplicity.

Despite the theoretically computational efficiency and parametric efficiency of group convolutions, they slow down the training in practice (Ma et al., 2018). To alleviate this issue, we provide another efficient version of EIT, namely E-EIT, by simplifying the design of dual enhanced interaction. As illustrated in Figure 3, both ISI and CSI adopt a single-layer operation. Formally, the dual enhanced interactions are computed as:

$$\ddot{\mathbf{S}} = \mathbf{g}^{(\mathbf{0})}(\mathbf{ReLU}(\mathbf{f}^{(\mathbf{0})}(\mathbf{S}))), \qquad (6)$$

where $\mathbf{ReLU}(\mathbf{f}^{(\mathbf{0})}(\mathbf{S}))$, namely as $\dot{\mathbf{S}}$, $\in \mathbb{R}^{M^H \times T \times T}$ and $\ddot{\mathbf{S}} \in \mathbb{R}^{M \times T \times T}$ and $M^H$ is a hyper-parameter, e.g. we set it as 32 for the base configuration. In this way, E-EIT avoids parts of memory consumption and somehow improves the computational efficiency.

## 4 Experiment Settings

We evaluate our EIT on five widely used benchmarks: 1) Machine Translation, 2) Grammar Error Correlation, 3) Abstractive Summarization, and 4) Language Modeling. The detailed architecture setups, training setups and evaluation setups are presented in Appendix A.

### 4.1 Machine Translation

**Dataset.** We select two widely used corpus: WMT'14 English-German (En-De) translations (a large-scale dataset, 4.5M training sentence pairs) and WMT'16 English-Romanian (En-Ro) translations (a small-scale dataset, 610K training sentence pairs). The validation and test sets are *newstest2013* and *newstest2014*, respectively. For the En-Ro task, it consists of 610K training sentence pairs. The preprocessing follows the setups in Mehta et al. (2021).

**Models.** We build our EIT vaiants of different configurations ranging over *base*, *big* and *deep* for both tasks. The configurations are the same as that in Vaswani et al. (2017).

### 4.2 Grammar Error Correlation

**Dataset.** We also examine the effectiveness of EIT on grammar error correction task, an important application in natural language processing. We conduct experiments on CONLL dataset, which consists of 827K training sentences. We replicate the setup in (Chollampatt and Ng, 2018) and adopt the word-level dropout technique (Sennrich et al., 2016) to alleviate the overfitting problem.

**Models.** We choose the Transformer (Vaswani et al., 2017) and SURFACE (Liu et al., 2021) as the comparisons. All architectures follow the Transformer-base configuration in Vaswani et al. (2017).

### 4.3 Abstractive Summarization

**Dataset.** We also test the effectiveness of EIT on abstractive summarization task, a task relying on the ability of modeling long dependency. We select a widely used corpus: CNN/DailyMail dataset which consists of 287K training documents.

**Models.** Our models are all under base configuration, e.g., embedding dimension, hidden dimension, $M$ are set to 512, 2048 and 8, respectively.

### 4.4 Language Modeling

**Dataset.** We also evaluate our EIT on a language modeling task: WikiText-103 to further examine its ability of modeling long-dependency. The training, validation and test sets consist of 103M words (from 28K articles), 218K words and 246K words, respectively. We follow the official preprocessing procedure (Ott et al., 2019)

**Models.** We select Adaptive Input Transformer (Baevski and Auli, 2019) as the baseline. Both the baseline and our EIT are all 8-layer big models with 8 heads.

## 5 Experiments Results

### 5.1 Machine Translation

**Performance.** Table 1 and Table 2 display the results on En-De and En-Ro tasks, respectively. First, we can see that Our EIT variants demonstrate superior performance compared to the vanilla Transformer across various configurations on both

| Type | Model | WMT'14 En-De | | |
|---|---|---|---|---|
| | | $\theta$ (M) | BLEU | sBLEU |
| Head Modification | Refiner (Zhou et al., 2021b) | - | 27.62 | - |
| | Talking-Head (Shazeer et al., 2020) | - | 27.51 | - |
| | Collaboration (Wang and Tu, 2020) | - | 27.55 | - |
| | DYROUTING (Li et al., 2019) | 297M | 28.96 | - |
| | DISAGREE (Li et al., 2018) | - | 29.28 | - |
| | MoA (Zhang et al., 2022) | 200M | 29.40 | - |
| | FISHformer (Nguyen et al., 2022) | - | 29.57 | - |
| Localness | DMAN (Fan et al., 2021) | 211M | 28.97 | 27.8 |
| | CSAN (Yang et al., 2019) | - | 28.74 | - |
| | UMST (Li et al., 2022) | 242M | 29.75 | - |
| Our System (Pre-Norm) | Transformer base | 62M | 27.13 | 26.0 |
| | EIT base | 62M | **28.00** | 26.9 |
| | E-EIT base | 62M | 27.72 | 26.7 |
| | Transformer 48L | 194M | 29.60 | 28.5 |
| | EIT 48L | 194M | **30.25** | 29.2 |
| | E-EIT 48L | 194M | 30.16 | 29.1 |
| | Transformer big | 211M | 28.80 | 27.7 |
| | EIT big | 212M | **29.79** | 28.7 |
| | E-EIT big | 211M | 29.61 | 28.5 |

Table 1: Results on WMT'14 En-De Task.

| Type | Model | WMT'16 En-Ro | |
|---|---|---|---|
| | | $\theta$ (M) | BLEU |
| Basic Baseline | Transformer (Liu et al., 2020) | - | 34.30 |
| | Transformer (Kasai et al., 2020) | | 34.16 |
| | DELIGHT (Mehta et al., 2021) | 53M | 34.70 |
| Head modification | Refiner (Zhou et al., 2021b) | 54M | 34.25 |
| | Talking-Head (Shazeer et al., 2020) | 54M | 34.35 |
| | Collaboration (Wang and Tu, 2020) | 54M | 34.64 |
| | FISHformer (Nguyen et al., 2022) | 49M | 34.42 |
| | MoA (Zhang et al., 2022) | 56M | 34.39 |
| Localness | DMAN (Fan et al., 2021) | - | 34.49 |
| | UMST (Li et al., 2022) | 60M | 34.81 |
| Our System (Pre-Norm) | Transformer base | 54M | 34.23 |
| | EIT base | 54M | **35.10** |
| | E-EIT base | 54M | 35.01 |
| | Transformer 24L | 111M | 35.00 |
| | EIT 24L | 111M | **35.40** |
| | E-EIT 24L | 111M | 35.35 |
| | Transformer big | 196M | 34.44 |
| | EIT big | 196M | **34.91** |
| | E-EIT big | 196M | 34.67 |

Table 2: Results on WMT'16 En-Ro Task.

tasks. This indicate the effectiveness of EIT variants. Notably, E-EIT, an alternative to satisfy the low-latency of industrial application, can deliver competitive results compared with the full version while maintaining fast processing speeds.

Besides, Our EIT can beat all selected methods of head modification and localness modeling, including the latest methods such as MoA (Zhang et al., 2022), Fishformer (Nguyen et al., 2022), UMST (Li et al., 2022), on both datasets. This highlights the fact that focusing on a single aspect, such as complementarity, is inadequate for achieving optimal results. It is essential to take into account both complementarity and consensus to ensure the best outcomes.

**Efficiency.** As the application of EIT is limited to the encoder-side only, the impact on inference speed degradation is insignificant.

### 5.2 Grammar Error Correlation

Table 3 presents the results on the CONLL dataset's test set. Both EIT and E-EIT outperform the standard Transformer, showing improvements of 0.87 and 1.13 in terms of $F_{0.5}$, respectively. Compared to the strong baseline SURFACE, our methods (EIT and E-EIT) still outperform it by 0.45 and 0.71 $F_{0.5}$ points, respectively. Importantly, both EIT and E-EIT require negligible extra parameters, less than 0.1M, indicating their enhanced expressive power.

### 5.3 Abstractive Summarization

Table 4 shows results on test set of CNN-DailyMail. We can see EIT can achieve scores of 41.62

ROUGE-1 points, 18.70 ROUGE-2 points and 38.33 ROUGE-L points, outperforming the standard Transformer by 0.78, 0.70 and 0.75 in terms of ROUGE-1, ROUGE-2 and ROUGE-L points, respectively. Compared with other strong baselines, our EIT can still show superiority on these datasets in terms of ROUGE-1 points, e.g., EIT surpasses SURFACE, DMAN and BOTTOM-UP by 0.62, 0.64 and 0.40 in terms of ROUGE-1 points, respectively. Notably, our efficient version of EIT, the E-EIT can achieve comparable performance with EIT.

### 5.4 Language Modeling

Table 5 presents the perplexity scores of various models on the WikiText-103 test set. Our EIT and E-EIT models outperform the baseline with PPL scores of 1.11 and 0.92, respectively. These results highlight the high expressiveness of our methods, as the improvements are achieved with only a negligible increase in parameters.

## 6 Analysis

### 6.1 Ablation Studies

Table 6 summarizes the impacts of removing each module on En-De and En-Ro tasks, respectively. First, we find removing any module (or sub-module) mostly results in obvious performance degradation (#3,4,5 vs. #2). These evidences indicate the indispensability of these modules.

Notably, when removing the M2M module (#2 vs. #3), we observe an obvious decline in performance on two translation tasks, indicating the

| Model | Precision | Recall | $F_{0.5}$ |
|---|---|---|---|
| Transformer ‡ | 64.84 | **36.61** | 56.18 |
| SURFACE (Liu et al., 2021) | 66.80 | 35.00 | 56.60 |
| EIT | **69.98** | 32.80 | 57.05 |
| E-EIT | 69.85 | 33.36 | **57.31** |

Table 3: Results on the correction task.

| Model | RG-1 | RG-2 | RG-L |
|---|---|---|---|
| Transformer ‡ | 40.84 | 18.00 | 37.58 |
| PG-Net (See et al., 2017) | 39.53 | 17.28 | 36.38 |
| MADY (Wang et al., 2021) | 40.72 | 17.90 | 37.21 |
| DMAN (Fan et al., 2021) | 40.98 | 18.29 | 37.88 |
| BOTTOM-UP (Gehrmann et al., 2018) | 41.22 | 18.68 | **38.34** |
| SURFACE (Liu et al., 2021) | 41.00 | 18.30 | 37.90 |
| EIT | **41.62** | **18.70** | 38.33 |
| E-EIT | 41.58 | 18.63 | 38.28 |

Table 4: Results on the summarization task.

| Model | Depth | $\theta$ (M) | Test PPL |
|---|---|---|---|
| Adaptive Transformer | 8 | 147M | 21.11 |
| EIT | 8 | 147M | **20.00** |
| E-EIT | 8 | 147M | 20.19 |

Table 5: Results on the WikiText-103 dataset.

| # Model | En-De | | En-Ro | |
|---|---|---|---|---|
| | BLEU | Time | BLEU | Time |
| 1 Transformer | 27.13 | - | 34.23 | - |
| 2 EIT | **28.00** | 1.45× | **35.10** | 1.38× |
| 3   - Many-to-Many | 27.39 | 1.15× | 34.71 | 1.10× |
| 4   - Inner-Subspace Interaction | 25.79 | 1.22× | 32.50 | 1.21× |
| 5   - Cross-Subspace Interaction | 27.70 | 1.40× | 34.53 | 1.29× |

Table 6: Ablation study on two tasks. Time denotes the training computing time.

importance of M2M module. Within our EIT framework, the M2M module, motivated by the *complementary principle*, serves the critical purpose of supplying necessary information for subsequent interactions. Therefore, its absence impedes the effectiveness of our two interaction models.

Furthermore, the omission of the ISI sub-module (#2 vs. #4) results in a significant and noticeable decrease in BLEU scores. One possible explanation is that while increasing the number of heads enhances the information capacity, it also introduces a certain degree of irrelevant information (noise) into the attention maps. Consequently, a direct fusion of these heads fails to yield satisfactory outcomes. However, our EIT framework overcomes this challenge by incorporating the ISI sub-module, which provides an effective mechanism for discarding irrelevant information while retaining the benefits of the previous heads. This unique and innovative design sets our approach apart from the *attention expansion* technique (Zhou et al., 2021b).

Apart from performance, we can see that the additional primary cost comes from the ISI sub-module (#2 vs. #4), which occupies nearly 50% extra training cost. We attribute this phenomenon to the unfriendly support for the implementation of group convolution in PyTorch (Paszke et al., 2019).

## 6.2 Analysis on Placement of DEI

Table 7 compares the impacts on several placements of DEI module, e.g., ISI→ Softmax →CSI. First, Softmax operation is insensitive to the placement of the CSI sub-module, which results in neg-ligible BLEU degradation (#1 vs. #2). Moreover, by comparing #2 and #3, we find that when placing the ISI sub-module behind the Softmax, the performance suffers an obvious BLEU drop on both tasks compared to EIT. One potential explanation for this phenomenon is that the softmax operation, which averages out the noise information to every position, can be detrimental to the ISI module.

## 6.3 Parameter Analysis

**Effect of $M$.** In our EIT, the size of $M$ directly influences the number of attention maps we can obtain, e.g., $M^2$ attention maps. To investigate its effect on performance, we conduct experiments with different $M$ on the En-De task. The results are exhibited in Figure 4(a). We can see that the performance gap between EIT and Transformer increases as the $M$ grows. This indicates that EIT has good ability of utilizing these attention maps. For example, when $M$ is 16, EIT achieves BLEU scores of 28.06 on the En-De task.

**Effect of Strength of M2M.** In our default setting of many-to-many mapping, each query can attend to all the keys. We further investigate the effect of strengths of M2M on the final performance. Strength refers to the number of keys each query can attend to. Figure 4(b) displays the results. We can see that as the strength increases, the performance on the En-De task goes better. This is reasonable since more strength the more information contains which is better for later interaction.

**Effect of Number of EIT Layers.** Recent researches (Shi et al., 2016; Peters et al., 2018; Hao et al., 2019) has demonstrated that various layers

| # | IB | IA | CB | CA | En-De | En-Ro |
|---|----|----|----|----|-------|-------|
| 1 | ✓ |   | ✓ |   | **28.00** | **35.10** |
| 2 | ✓ |   |   | ✓ | 27.93 | 34.88 |
| 3 |   | ✓ |   | ✓ | 27.50 | 34.45 |

Table 7: Effect on placement of DEI on two tasks. Here, IB and IA denote the ISI module is located *before* or *after* the Softmax function, and so on for CSI.

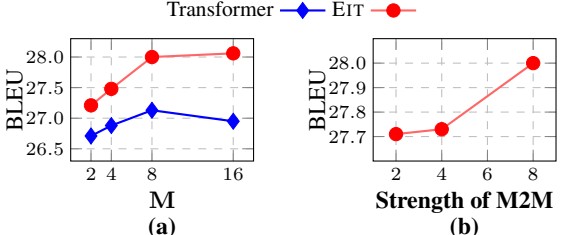

Figure 4: BLEU vs. the size of $M$ and strength of M2M on the En-De task.

| Encoder Layers | Training Time | En-De |
|----------------|---------------|-------|
| [1] | 1.07× | 27.76 |
| [2] | - | 27.46 |
| [3] | - | 27.40 |
| [4] | - | 27.38 |
| [5] | - | 27.30 |
| [6] | - | 27.48 |
| [1 − 2] | 1.13× | **28.08** |
| [1 − 3] | 1.20× | 28.02 |
| [1 − 4] | 1.27× | 28.05 |
| [1 − 5] | 1.36× | 27.82 |
| [1 − 6] | 1.45× | 28.00 |

Table 8: Layer Evaluation of Encoder with EMHA Implementation. "1" indicates the bottom layer.

in the encoder of a model have a tendency to capture distinct syntax and semantic features. Consequently, each layer may have different requirements for promoting agreement among the representations. In light of this, we examine the impact of consensus on different layers. The results on the En-De task are presented in Tables 8. The lowest layer clearly benefits from a higher degree of consensus compared to other layers, consistent with prior research (Shleifer and Ott, 2022) indicating the challenges of optimizing shallow layers within the pre-normalization paradigm. However, by employing the consensus strategy, we enhance the learning of representations in shallow layers, giving them a significant advantage. Additionally, it is observed that incorporating consensus into multiple layers does not yield optimal results. These findings suggest an efficient strategy: apply EMHA exclusively to the first encoder layer of EIT for optimal efficiency.

### 6.4 Analysis on Behaviour of Attention Heads

#### 6.4.1 EIT owns Higher Consensus among Attention Heads

As depicted in Figure 5, it is evident that EIT exhibits the highest average similarity among attention maps from various heads, surpassing all other models. This finding suggests that EIT demonstrates a greater consensus among attention heads. We attribute this achievement to the significant role played by M2M and dual enhanced interaction. M2M facilitates the generation of rich information, while dual enhanced interaction efficiently leverages and refines the available information from different attention heads.

**Discussions.** This phenomenon is contradictory to the findings of previous studies about head interaction (Wang et al., 2022a). We speculate that this is because our interactions are more efficient, not only relying on an adequate number of attention heads but also operating in a hierarchical manner. These characteristics result in a consensus among the attention maps.

#### 6.4.2 Dynamics of Attention Map Similarity during Computation

Figure 6 exhibits the dynamics of attention map similarity for the EIT 48L model on the En-De test set. The similarity between attention maps initially decreases and then increases as the dual interactions progress. This pattern is attributed to the two stages of our approach. In the ISI phase, interactions are modeled within each group, generating representative attention maps. As these groups operate independently, the similarity among these representatives is lower. Subsequently, in the CSI phase, interactions occur among these representatives, resulting in the final attention maps. This CSI enhances similarity among the attention maps.

#### 6.4.3 EIT Learns High-quality Representations

We further investigate how consensus affect the layer representations. Following (Gong et al., 2021; Dong et al., 2021; Shi et al., 2022; Wang et al., 2022b), we adopt the token correlation $\mathcal{TC}$ to measure the quality of features (the lower, the better). The token correlation is computed by the Pearson correlation coefficient (Benesty et al., 2009).

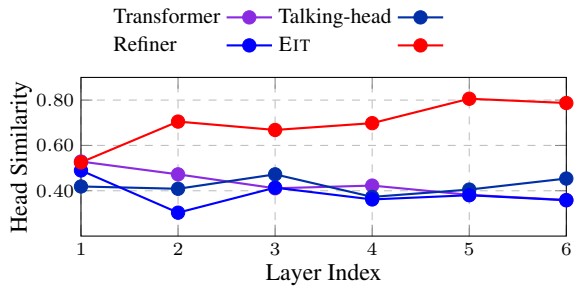

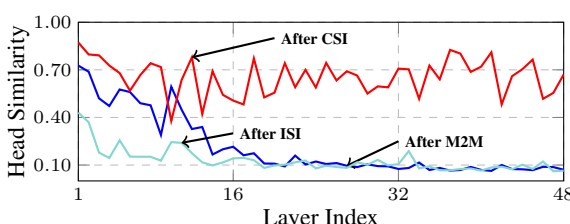

Figure 5: Cosine similarity among attention maps of different models on En-De task.

Figure 6: Dynamics of attention map similarity.

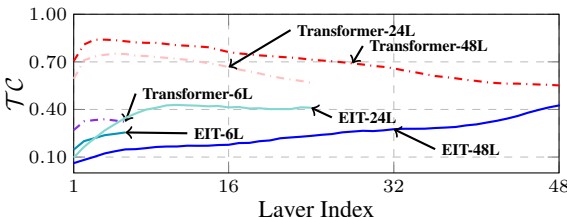

Figure 7: Token correlation of Transformer and EIT on En-De task (Left) and CNN-DailyMail task (Right).

| Model | Pruning Ratio | | |
|---|---|---|---|
| | 0.0% | 50.0% | 87.5% |
| Transformer-48L | 29.60 | 27.64 | 1.86 |
| EIT-48L | **30.25** | **29.09** | **21.12** |

Table 9: BLEU points of models with head pruning on the En-De task.

Figure 7 exhibits the results on the test set of the En-De task. Notably, the features learned by EIT exhibit lower token correlation compared to the vanilla Transformer across all configurations. This indicates that EIT effectively learns improved layer representations.

Furthermore, we observe that the vanilla Transformer consistently maintains relatively high token correlation from the first layer. This observation aligns with prior study (Shleifer and Ott, 2022), suggesting that lower layers struggle to optimize effectively in pre-normalization Transformers. However, our EIT approach alleviates this issue.

### 6.4.4 EIT Makes Head Pruning Easier

To further explore the possibility of pruning the consensus attention maps, we introduce a simple head mask mechanism for head pruning during the inference phase as follows:: $\mathbf{O} = \sum_{i=1}^{M} \eta_i \mathbf{A}^i \mathbf{X} \mathbf{W}_V^i \mathbf{W}_O^i$, where $\eta_i \in \{0, 1\}$. Table 9 exhibits the results on En-De tasks. Note that the head selection process is done in a straightforward manner, such as selecting heads by index, without considering their relative importance as highlighted in previous studies (Michel et al., 2019). Additionally, the head pruning operations are exclusively applied to the encoder side. It is evident that EIT exhibits a high tolerance for head pruning without experiencing significant deterioration in performance. Such phenomenon sheds light on the researches of head pruning and inference speeding.

## 7 Related Work

**Improved Multi-Head Mechanism** Previous work has shown that multi-head attention can be further enhanced by encouraging individual attention heads to extract distinct information (Li et al., 2018; Cui et al., 2019; Sukhbaatar et al., 2019; Guo et al., 2020; Hao et al., 2019). Another branch of research is designing more complex interactive modeling to make better use of the multiple subspace information (Shazeer et al., 2020; Wang and Tu, 2020; Li et al., 2019). Besides, Voita et al. (2019) empirically demonstrates that some heads in attention are useless and can be pruned without performance degradation. Along this line, researchers investigate how to efficiently cut off redundant heads (Michel et al., 2019; Behnke and Heafield, 2020). Different from these work, our study aims to leverage the benefits of both diversity and consistency.

## 8 Conclusions

In this paper, we propose EIT, an alternative to the Transformer architecture. It further advances the multi-head schema by fully leveraging two principles in multi-view learning: the *complementary principle* and the *consensus principle*. In addition, E-EIT can be served as another choice considering the trade-off between performance and computation efficiency. Experimental results on four widely-used tasks demonstrate the effectiveness of EIT-variants, which deliver consistent improvements to the standard Transformer.

## Limitations

Besides the advantages endowed by EIT, there still exists a shortcoming that the computational efficiency of the group convolution cannot be satisfactory, although it is computationally efficient in theory. This is due to the lack of high-efficiency CUDA kernel support. We will release a more efficient optimization of group convolutions in the soon future.

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

## A   Detailed Setups of Experiments

### A.1   Machine Translation Task

**Dataset**   We evaluated our approach on two widely used machine translation datasets: WMT'14 En-De and WMT'16 En-Ro. The En-De dataset contains approximately 4.5M tokenized training sentence pairs. We selected newstest2013 and newstest2014 as the validation and test data, respectively. As for the En-Ro dataset, it consists of 0.6M tokenized training sentence pairs. We performed shared BPE operations on both datasets to overcome the out-of-vocabulary (OOV) problem. Concretely, we set the size of BPE operations to 32K and 20K for En-De and En-Ro datasets, resulting in a shared vocabulary with sizes of 34040 and 19064, respectively.

**Model Configuration**   Our model architectures are based on Transformer (Vaswani et al., 2017). We provided three basic configurations, namely *base*, *deep*, and *big* which follow the configurations in Vaswani et al. (2017). We adopted a pre-normalization strategy (Wang et al., 2019) considering training stability under different configurations.

The detailed settings of hyper-parameters are given in Table 11.

**Training & Evaluation**  Our implementations are based on Fairseq (Ott et al., 2019). Our experiments are performed on the GEFORCE RTX 3090 cards. We use 8 GEFORCE RTX 3090 cards to train models for the WMT'14 En-De task. As for the models on the WMT'16 En-Ro task, we train them on 4 GEFORCE RTX 3090 cards. The batch sizes for En-De and En-Ro tasks are 65536 and 16384, respectively. The total updates are 50K, 50K and 100K for *base*, *deep* and *big* in En-De task, respectively. We adopt Adam (Kingma and Ba, 2015) as an optimizer with an $\text{adam}_\beta$ of (0.9, 0.997). The learning rate scheduler is *invert sqrt* with a learning rate of 0.002 and warmup updates of 16000. We also adopt label smoothing with a ratio of 0.1 in all the experiments. More details are exhibited in Table 12. During the evaluation process, we set the beam number to 4 and the length penalty to 0.6 for the En-De task. As for the En-Ro task, the number of beams is 5 and the length penalty is 1.3.

### A.2  Abstractive Summarization Task

**Dataset**  For abstractive summarization, we conduct experiments on a widely used corpus, e.g., CNN/DailyMail dataset. It consists of 287K training documents. Shared BPE operations with a size of 30K are performed on all the training data, resulting in a vocabulary of 32584.

**Model Configuration**  We only provide the *base* configuration of our EIT and E-EIT for abstractive summarization. The details are presented in Table 11.

**Training & Evaluation**  We train models for an abstractive summarization task on 8 GEFORCE RTX 3090 cards with a batch size of 131072 and total updates of 30K. We adopt a weight decay strategy with a ratio of 0.0001. Other hyper-parameters are the same as that in machine translation tasks. You can find their settings in Table 12. During testing, the number of beams is set to 4 and the length penalty is set to 2.0. Besides, we set the minimal length and maximum length to 55 and 140, respectively.

### A.3  Grammar Error Correction Task

**Dataset**  For the grammar error correction task, we select the CONLL dataset to evaluate our approach. The CONLL dataset consists of 827K training sentences. We replicate the setup in Chollampatt and Ng (2018) and adopt the word-level dropout technique (Sennrich et al., 2016) to alleviate the overfitting problem. More details are listed in Table 10.

**Model Configuration**  For grammar error correction task, we only provide the *base* configuration of our EIT and E-EIT. The details are presented in Table 11. Notice that the models on this task adopt a post-normalization strategy.

**Training & Evaluation**  We train models for the grammar error correction task on 8 GEFORCE RTX 3090 cards. The batch size is 65536 and the total updates are 14K. More training details are shown in Table 12. During testing, the beams and length penalty are set to 6 and 0.6, respectively.

### A.4  Automatic Disease Diagnosis Task

**Dataset**  For the automatic disease diagnosis task, we select the ABIDE dataset to evaluate our approach. The ABIDE dataset consists of 1009 brain networks from 1009 real samples of 17 international sits. Due to the heterogeneity of this data, we adopt the shared data with re-standardized data splitting in Kan et al. (2022). Specifically, 70%, 10% and 20% samples are served as the training, validation and test sets, respectively.

**Model Configuration**  For ABIDE task, we still follow the model configuration in Kan et al. (2022). Specifically, we build our BrainNetEITF with two-layer encoder. The number of heads $M$ are set to 4 for each layer.

**Training & Evaluation**  We train all models including the BrainNetTF and BrainNetEITF fro 200 epochs on a single GEFORCE RTX 3090 card. Each model is trained by 5 times. We adopt Adam (Kingma and Ba, 2015) as an optimizer with an initial learning rate of $10^{-4}$ and a weight decay of $10^{-4}$. The batch size is set to 64. We adopt the checkpoint of the final epoch for evaluating the test set.

### A.5  Language Modeling Task

**Dataset**  For the language modeling task, we select the WikiText-103 dataset to evaluate our approach. The training set consists of 103M words from 28K articles. While for the validation and test sets, they are made up of 218K and 246K words, respectively. In details, we follow the instructions in

| Dataset | Sentence | | | BPE | Vocab |
|---|---|---|---|---|---|
| | **Train** | **Dev** | **Test** | | |
| WMT'14 En-De | 4.5M | 3.0K | 3.0K | 32K | 34040 |
| WMT'16 En-Ro | 0.6M | 2.0K | 2.0K | 20K | 19064 |
| CNN/DailyMail | 287K | 13.0K | 11.0K | 30K | 32584 |
| CONLL | 827K | 5.4K | 1.3K | 30K | 33136 |
| WikiText-103 | 103M | 218K | 246K | - | 267740 |

Table 10: The details of datasets of language tasks.

| Task | Model | Configuration | $M$ | $M^H$ | $M^{H_{isi}}$ | $M^{H_{csi}}$ | $r$ | $K_h^{isi}$ | $K_w^{isi}$ | $K_h^{csi}$ | $K_w^{csi}$ |
|---|---|---|---|---|---|---|---|---|---|---|---|
| MT | EIT | *base* | 8 | - | 128 | 64 | 8 | 1 | 7 | 1 | 3 |
| | | *deep* | 8 | - | 128 | 64 | 8 | 1 | 7 | 1 | 3 |
| | | *big* | 16 | - | 256 | 256 | 16 | 1 | 7 | 1 | 3 |
| | E-EIT | *base* | 8 | 32 | - | - | 8 | 1 | 7 | 1 | 7 |
| | | *deep* | 8 | 32 | - | - | 8 | 1 | 7 | 1 | 7 |
| | | *big* | 16 | 64 | - | - | 16 | 1 | 7 | 1 | 7 |
| AS | EIT | *base* | 8 | - | 8 | 64 | 8 | 1 | 1 | 1 | 1 |
| | E-EIT | *base* | 8 | 16 | - | - | 8 | 1 | 1 | 1 | 1 |
| GEC | EIT | *base* | 8 | - | 128 | 128 | 8 | 1 | 7 | 1 | 3 |
| | E-EIT | *base* | 8 | 64 | - | - | 8 | 1 | 7 | 1 | 7 |
| LM | EIT | *big* | 8 | - | 64 | 32 | 8 | 1 | 1 | 1 | 1 |
| | E-EIT | *big* | 8 | 8 | - | - | 8 | 1 | 1 | 1 | 1 |

Table 11: The configurations of models on three sequence generation tasks. MT, AS, GEC and LM denote machine translation, abstractive summarization, grammar error correction and language modelling, respectively.

| Hyper-parameter | WMT'14 En-De | WMT'16 En-Ro | CNN/DailyMail | CONLL | WikiText-103 |
|---|---|---|---|---|---|
| GPUs | 8 | 4 | 8 | 8 | 8 |
| Batch | 4096 | 4096 | 4096 | 4096 | 1024 |
| UF | 2 | 1 | 4 | 2 | 8 |
| Optimer | Adam | Adam | Adam | Adam | Nag |
| $\text{Adam}_\beta$ | (0.9, 0.997) | (0.9, 0.997) | (0.9, 0.997) | (0.9, 0.980) | - |
| LR | 0.0020 | 0.0020 | 0.0020 | 0.0015 | 0.0001 |
| LR scheduler | inverse sqrt | inverse sqrt | inverse sqrt | inverse sqrt | Cosine(t-mult=2) |
| Initial LR | $1e^{-7}$ | $1e^{-7}$ | $1e^{-7}$ | $1e^{-7}$ | $1e^{-7}$ |
| Total updates | 50K (100K) | 25K | 30K | 14K | 286K |
| Warmup updates | 16000 | 8000 | 8000 | 4000 | 16000 |
| Weight decay | 0.0000 | 0.0000 | 0.0001 | 0.0001 | 0.0000 |
| Label smoothing | 0.1 | 0.1 | 0.1 | 0.1 | 0.0 |
| Dropout | 0.1 (0.3) | 0.1 (0.3) | 0.1 | 0.2 | 0.3 |
| Attention dropout | 0.1 | 0.1 | 0.1 | 0.1 | 0.1 |
| ReLU dropout | 0.1 | 0.1 | 0.1 | 0.1 | 0.1 |
| Word dropout | 0.0 | 0.0 | 0.0 | 0.2 | 0.1 |

Table 12: The training setups of different tasks. UF denotes the update frequency of the gradient. (.) lists the values of hyper-parameters under the *big* configuration, which vary from the values under the *base* configuration.

Fairseq (Ott et al., 2019) to obtain and preprocess the data. The details are listed in Table 10.

**Model Configuration** For WikiText-103 task, Both baseline and our model are all 8-layer big model with 8 heads. Note that the baseline we

adopted are adaptive input transformer (Baevski and Auli, 2019). In this task, the kernel sizes in DEI are all set to 1.

**Training & Evaluation** The training and evaluation settings all follow the standard instructions for language modeling in PyTorch (Ott et al., 2019). We train both baseline and EIT with 286000 updates. The details are given in Table 12. As for the evaluation process, we adopt the checkpoint performing best on the validation set. We set the max-tokens, max-sentences, context-window to 3072, 1 and 2560, respectively.

## B Details of Metrics

### B.1 Calculation of Head Distance

Inspired by the attention metrics in Zhou et al. (2021a) and Wang et al. (2022b), we measure the distance between different heads by calculating cosine similarity among attention maps. Notice that our metric focuses on the diversity of attention maps, which is quite different from them. Denote the dataset as $\mathcal{D}$, and the attention map of $h$-th head of $l$-th layer of $i$-th sample denotes as $\mathbf{A}^{(h,l,i)}$, the head similarity in $l$-th layer is computed by averaging the cosine similarity of every two heads in $i$-th layer across all samples as:

$$
\begin{aligned}
\mathcal{HD}^{(l)} = & \frac{1}{|\mathcal{D}|}\frac{1}{M(M-1)} \\
& \frac{1}{T}\sum_{i=1}^{|\mathcal{D}|} \\
& (\sum_{j=1}^{M}\sum_{k=1}^{M} \\
& \sum_{t=1}^{T}\text{Cosine}(\mathbf{A}_{t,:}^{(j,l,i)},\mathbf{A}_{t,:}^{(k,l,i)}) \\
& - M)
\end{aligned} \tag{7}
$$

where $|\mathcal{D}|$ denotes the size of dataset, $M$ is the number of partition of features in attention, $T$ is the sequence length and $\text{Cosine}(\cdot)$ denotes the cosine similarity function. We set $\mathcal{D}$ to the test set of the corresponding task. The obtained head similarity ranges from [0, 1]. The larger the head similarity, the lower the distances between different heads are.

### B.2 Calculation of Token Correlation

We define a metric $\mathcal{TC}$, which measures the correlation among the representations of different tokens.

Denote the dataset as $\mathcal{D}$, and the sequence representation of $i$-th sample in $l$-th layer denotes as $\mathbf{X}^{(l,i)}$, the token correlation of in $l$-th layer is computed as:

$$
\begin{aligned}
\mathcal{TC}^{(l)} = & \frac{1}{|\mathcal{D}|}\frac{1}{T(T-1)}\sum_{i=1}^{|\mathcal{D}|} \\
& (\sum_{j=1}^{T}\sum_{k=1}^{T}\rho(\mathbf{X}_j^{(l,i)},\mathbf{X}_k^{(l,i)}) - T)
\end{aligned} \tag{8}
$$

where $\rho(\cdot)$ denotes the pearson correlation function. Intuitively, the larger the $\mathcal{TC}$ is, the higher the token correlation is, degrading the model's learning capacity (Gong et al., 2021).

## C Detailed added parameters of our methods

The detailed parameters of models on all tasks are listed in Table 13 and Table 14. We can see that the increased parameters are negligible on all tasks. Thus, we can exclude the effect of increasing parameters on performance.

### C.1 Efficiency Comparison

Despite the performance evaluation, the memory consumption and computational cost are also two major concerns in the literature. Figure 8 also displays the memory consumption and computational cost of models on the En-De task. EIT only costs 8.5% more memory consumption and 44.4% more training costs than the baseline with a depth of 6. However, the extra consumption goes larger as the depth goes deeper.

Besides, as aforementioned, we elaborately design an efficient version E-Eit that only costs 9.4% more memory consumption and 21.7% more training costs than the baseline under all the configurations on average. In this work, the many-to-many mapping rule is only applied on the encoder side. This is because the proposed M2M module and the subsequent ISI and CSI sub-modules will significantly enlarge the inference cost due to the heavy use of product attention on the decoder side, although it can attain further benefits in terms of BLEU.

## D Visualization of Training and Validation Perplexity

We plot the training and validation perplexity of Transformer and our EIT on the WMT'14 task in

| Model | En-De | | | En-Ro | | |
|---|---|---|---|---|---|---|
| | Base | Deep-48L | Big | Base | Deep-24L | Big |
| Transformer | 61.56 M | 193.96 M | 211.22 M | 53.90 M | 110.64 M | 195.88 M |
| EIT | 61.63 M | 194.32 M | 211.55 M | 53.98 M | 111.09 M | 196.40 M |
| E-EIT | 61.57 M | 194.14 M | 211.30 M | 53.92 M | 110.73 M | 195.97 M |

Table 13: Detailed parameters of models on WMT En-De and WMT En-Ro tasks.

| Model | CNN-DailyMail | CONLL | WikiText-103 | ABIDE |
|---|---|---|---|---|
| Transformer | 60.82 M | 61.10 M | 146.49 M | 3.98 M |
| EIT | 60.83 M | 61.19 M | 146.50 M | 3.98 M |
| E-EIT | 60.82 M | 61.15 M | 146.49 M | 3.98 M |

Table 14: Detailed parameters of models on CNN-DailyMail, CONLL, WikiText-103 and ABIDE tasks.

Figure 9. We can see that our EIT owns lower training and validation perplexity than Transformer.

## E Hyper-Parameters Analysis (Kernel Size and Hidden Size)

Since there are several hyper-parameters in both ISI and CSI sub-modules, it is necessary to figure out how they affect performance. Figure 10 (a-d) plots the performance of EIT against the kernel size and the hidden size. We can see that EIT can outperform Transformer in all choice of kernel size and hidden size. This observation can further help us trade off efficiency and performance well. For example, we can set csi kernel size to 1 or isi kernel size to 3 or $M^{H_{isi}}$ to $M^2$ or $M^{H_{csi}}$ to $4M$ to own a more efficient EIT.

## F Local Analysis

Local modeling is one of the widely accepted ways to improve the expressiveness of Transformer (Yang et al., 2019; Fan et al., 2021; Li et al., 2022). In dual enhanced interaction, we apply convolution operations to attention maps, which has the potential to introduce local biases. To figure it out, we measure the localness of attention maps since if there is a local bias, each token will distribute larger attention weights on their neighboring tokens. We adopt the localness metric of Fan et al. (2021), denoted as $\mathcal{C}$ (higher is better). More details are presented in Appendix.

We plot the $\mathcal{C}$ value within a local region $w = 0.1 * T + 1$, of models in En-De task and CNN-DailyMail task in Figure 11. The value is computed over the test set. Due to the long sequence length, we only use a subset of the test set consisting of 1000 samples for the CNN-DailyMail task. The results (mean) show no significant local enhancement phenomena in both tasks. Note that the attention maps in the first layer of EIT on the abstractive summarization have a strong local pattern, but the kernel sizes are set to 1 on this task. So we conclude that the improvements do not come from local enhancement.

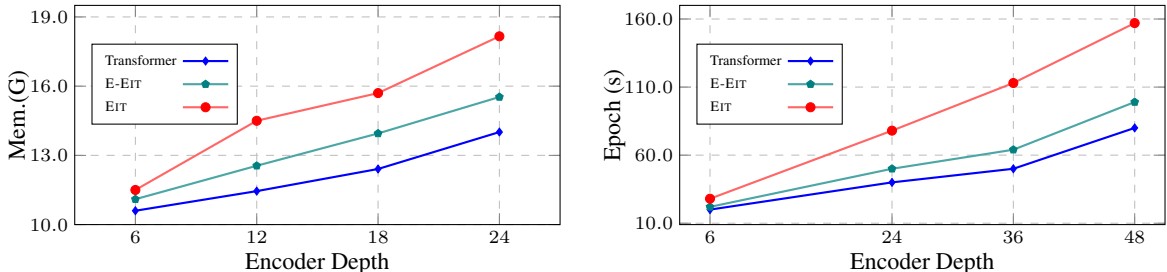

Figure 8: Memory and speed vs. encoder depth. E-EIT can achieve comparable results with fewer training costs than EIT.

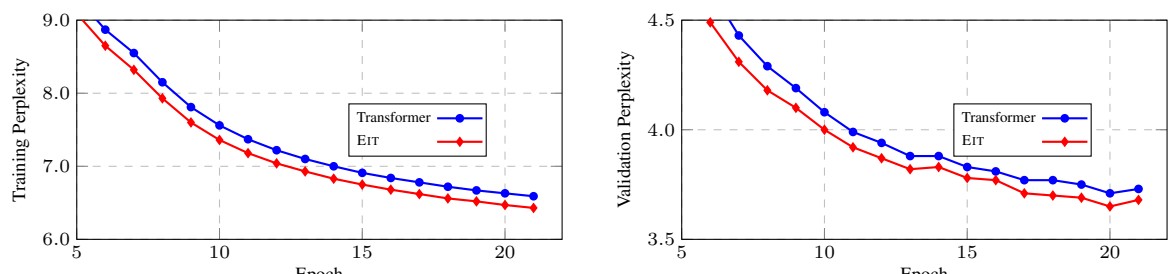

Figure 9: Training perplexity and validation perplexity of Transformer and our EIT on WMT'14 En-De task. Note that the models are in *base* configuration.

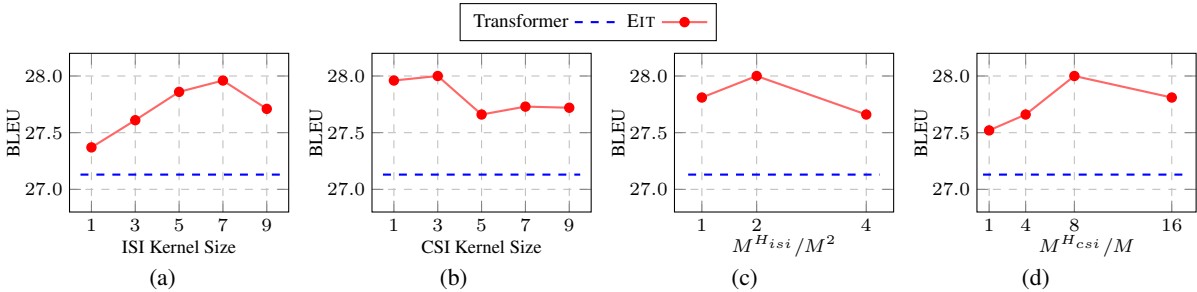

Figure 10: The comparison of BLEU against different hyper-parameters. Note that the blue horizontal line represents the performance of Transformer.

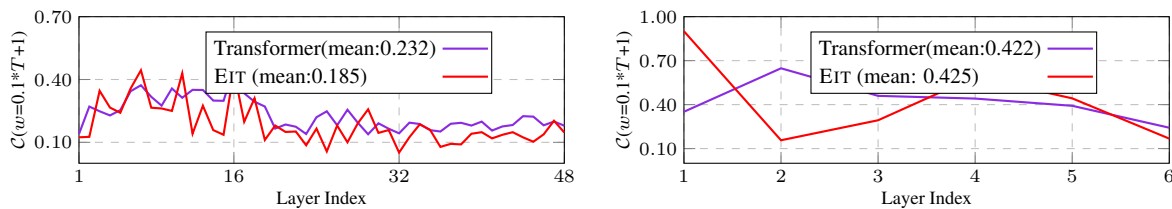

Figure 11: Quantitative analysis on localness in attention maps on En-De task (Above) and CNN-DailyMail task (Bottom).

