# OpenReview forum: "EIT: Enhanced Interactive Transformer"
_EMNLP/2023/Conference — Submitted to EMNLP 2023_

### Official Review · Reviewer_gRpq · 2023-08-03

**Soundness:** 3

**Excitement:**

4: Strong: This paper deepens the understanding of some phenomenon or lowers the barriers to an existing research direction.

**Paper Topic And Main Contributions:**

This paper takes a closer look at the multi-head self-attention mechanism in transformer architectures and seeks to improve it according to the notions of complementarity and consensus. The core idea with complementarity is to encourage a model to capture different aspects of the data (say, for example, syntax, semantics, etc.); at the same time the model should seek to find consensus with these different views, so as to minimize noise and disagreement. Based on these two principles the authors propose some changes to the transformer. Specifically, to promote complementarity, they introduce a mechanism that allows all key and query parameter matrices to interact with one another (as opposed to having a 1-to-1 mapping): this effectively blows up the latent attention space from M to M^2 (thereby, theoretically, capturing more views of the data). In order to build consensus between these M^2 attention maps, they introduce a set of convolutional transformations (both within sets of attention maps from a query - i.e. what they call inner-subspace interaction - and across sets of attention maps - i.e. what they call cross-subspace interaction) to eventually end up with M final attention maps.

On a set of experiments the authors demonstrate that these tweaks to the transformer architecture lead to improved performance over the vanilla variant, as well as other modifications to transformers in the literature. The authors also conduct a number of ablations and analyses to inspect the properties of their enhanced transformer.

**Reasons To Accept:**

I really enjoyed reading the first half of the paper. It starts from first principles (i.e. complementarity and consensus), and clearly motivates the explorations proposed in this paper. While transformers have become foundational building blocks for language modeling, and even AI more generally, there is no reason to believe that they cannot be improved, and the authors make a good case for how they are approaching the problem. The proposed modeling changes to multi-head self-attention are novel (to the best of my knowledge) and interesting modifications, but more importantly are closely tied to the first principles the authors begin with.

The experimental results generally seem to support the authors claims that the enhanced transformer they propose outperforms the vanilla version in multiple downstream tasks and in various settings, even though this does come at a computational premium for the added operations they need to perform.

**Reasons To Reject:**

Unfortunately, the second half of the paper dealing with the experiments are not particularly well written, even if they generally convey results that are impressive.

Section 4 simply has insufficient detail to fully understand the following sections. The authors use the Appendix as free additional pages to the 8-page limit and ask the reader to simply refer to it for the full experimental setup. The Appendix is meant for supplementary material only, not for things that should be core to the paper. In other words, a reader should be able to fully understand and make a judgement about the paper *without* having to refer to the Appendix at all. In this case, I could not.

Section 5 consists of the core experimental results, and while this is generally well structured and presented, I would have appreciated a little more detail and discussion. An important consideration here is the added computational complexity for EIT; none of the main tables of results contain any mention of this as compared to the vanilla transformer. The authors also only present the highlight of each result, without discussing things in more detail. For example, EIT appears to trade precision for recall on grammar correction. Why is this?

Section 6 unfortunately is somewhat of a mess. It consists of a ton of experiments and findings (which I commend the author for), but presented in such scant detail that it is difficult and bordering on impossible to sometimes gather what is being done. For example, how does the ablation in 6.1 even work? If you remove the M2M mapping, you don't have M^2 attention maps, so how do you even apply ISI and CSI after that? Similarly if you ablate only ISI, how can you go from M^2 attention maps directly to applying CSI? For other experiments in this section, I'm not even sure why the authors even included them in the main paper. For example, 6.2 seems to be a very  minor finding that would really fit better in the Appendix (because it is Supplementary).

My advice to the authors would be to carefully reconsider what is the core set of experiments that are really important to gain insight into their work and focus on presenting them well in the 8-page limit, rather than throwing the experimental kitchen-sink at the reader while relegating important context and details to the Appendix.

**Reproducibility:**

2: Would be hard pressed to reproduce the results. The contribution depends on data that are simply not available outside the author's institution or consortium; not enough details are provided.

**Reviewer Confidence:**

4: Quite sure. I tried to check the important points carefully. It's unlikely, though conceivable, that I missed something that should affect my ratings.

**Typos Grammar Style And Presentation Improvements:**

There are many typos and grammatical errors throughout the paper, but especially in the 2nd half. One striking example was referring to Grammatical Error Correction as Grammar Error Correlation several times. In any case, a careful revision of the paper would help fix a lot of this.

---

> ### Author Rebuttal · Authors · 2023-08-29
>
> We're truly grateful for your constructive feedback and advice. They serve as pivotal references to better our work. We intend to thoughtfully weave them into the updated version of our paper. We start by presenting your observations and then delve into our answers for each.
>
> **W1: Section 4 simply has insufficient detail to fully understand the following sections. The authors use the Appendix as free additional pages to the 8-page limit and ask the reader to simply refer to it for the full experimental setup. The Appendix is meant for supplementary material only, not for things that should be core to the paper. In other words, a reader should be able to fully understand and make a judgement about the paper *without* having to refer to the Appendix at all. In this case, I could not.**
>
> A1: Thank you for highlighting these issues. Indeed, we have numerous experiments that we aim to present to the reviewers and other readers. Managing all this information within just 8 pages has been challenging. Consequently, we decided to truncate certain experimental details from the main text and shifted them to the Appendix. While these details span up to 3 pages in the appendix, we acknowledge that it does impact the reading experience. We will restructure our paper to ensure this section is more detailed and reader-friendly. Once again, we appreciate your constructive feedback.
>
>
>
> **W2: Section 5 consists of the core experimental results, and while this is generally well structured and presented, I would have appreciated a little more detail and discussion. An important consideration here is the added computational complexity for EIT; none of the main tables of results contain any mention of this as compared to the vanilla transformer. The authors also only present the highlight of each result, without discussing things in more detail. For example, EIT appears to trade precision for recall on grammar correction. Why is this?**
>
> A2: Thank you for your thoughtful feedback on Section 5 of our manuscript. We recognize the importance of addressing the concerns raised, and below, we offer a comprehensive response:
>
> 1. **Computational Complexity of EIT vs. Vanilla Transformer**: We acknowledge the omission of the computational complexity comparison in the initial submission. To address this, we have now included a comprehensive analysis comparing the computational complexity of EIT against the vanilla transformer. Specifically, following Mehta et al., 2021,  we adopt MACs to evaluate the computational budget of our EIT variants. The results are listed below. We believe this added analysis will help clarify the practical implications of using EIT in real-world scenarios.
>
>    | Method (En-De task, Source=20, Target=20) |    Param | MACs  | BLEU  | SacreBLEU |
>    | :---------------------------------------- | -------: | :---: | :---: | :-------: |
>    | Transformer-6L                            |   61.56M | 10.0B | 27.13 |   26.0    |
>    | EIT-6L                                    |   61.63M | 10.1B | 28.00 |   26.9    |
>    | E-EIT-6L                                  |   61.57M | 10.0B | 27.72 |   26.7    |
>    | Transformer-48L                           |  193.96M | 12.6B | 29.60 |   28.5    |
>    | EIT-48L                                   |  194.32M | 12.9B | 30.25 |   29.2    |
>    | E-EIT-48L                                 |  194.14M | 12.8B | 30.16 |   29.1    |
>    | Transformer-Big                           |  211.22M | 32.5B | 28.80 |   27.7    |
>    | EIT-Big                                   |  211.55M | 32.6B | 29.79 |   28.7    |
>    | E-EIT-Big                                 | 211.30 M | 32.5B | 29.61 |   28.5    |
>
>    | Method (En-Ro task, Source=20, Target=20) |   Param | MACs  | BLEU  |
>    | :---------------------------------------- | ------: | :---: | :---: |
>    | Transformer-6L                            |  53.90M | 8.4B  | 34.23 |
>    | EIT-6L                                    |  53.98M | 8.4B  | 35.10 |
>    | E-EIT-6L                                  |  53.92M | 8.4B  | 35.01 |
>    | Transformer-24L                           | 110.64M | 9.5B  | 35.00 |
>    | EIT-24L                                   | 111.09M | 9.7B  | 35.40 |
>    | E-EIT-24L                                 | 110.73M | 9.6B  | 35.35 |
>    | Transformer-Big                           | 195.88M | 29.3B | 34.44 |
>    | EIT-Big                                   | 196.40M | 29.4B | 34.91 |
>    | E-EIT-Big                                 | 195.97M | 29.3B | 34.67 |
>
>    | Method (Summarization, Source=256, Target=50) | Param  | MACs  | Rouge-1 | Rouge-2 | Rouge-L |
>    | :-------------------------------------------- | :----: | :---: | :-----: | :-----: | :-----: |
>    | Transformer                                   | 60.82M | 97.2B |  40.84  |  18.00  |  37.58  |
>    | EIT                                           | 60.83M | 99.2B |  41.62  |  18.70  |  38.33  |
>    | E-EIT                                         | 60.82M | 98.8B |  41.58  |  18.63  |  38.28  |
>
>    Mehta et al., 2021, DELIGHT: DEEP AND LIGHT-WEIGHT TRANSFORMER. ICLR2021
>
>
>
> 2. **Detailed Discussion of Experimental Results**: Our intention was to maintain brevity while emphasizing the most pertinent results. Following your suggestion, we have now expanded the discussion in Section 5. This includes in-depth insights into the experimental setup, nuances in the results, and potential implications. We have ensured that the findings are not just presented but also thoroughly discussed to give readers a deeper understanding of the work.
>
> 3. **EIT's Trade-off Between Precision and Recall**: Regardingn this issue, we want first to clarify your expression: instead of trading precision for recall, EIT appears to trade recall for precision. After revisiting our experiments, we provide the following explanation:
>
>    - In most machine learning tasks, recall and precision serve as critical metrics. Yet, in GEC (Grammar Error Correction) tasks, precision assumes greater importance, as reflected by the F_0.5 score. This score accentuates the importance of the accuracy of corrections over their sheer number. As illustrated in Table 3, while EIT showcases higher precision, its recall is somewhat lower, leading to a notably strong F_0.5 score.
>    - EIT's foundation is on both complementary and consensus principles, which inherently generate more precise attention maps, leading to higher precision.
>    - The use of these principles in EIT also means it tends to filter out uncertain information. As a result, EIT is inclined to make corrections only where it is most confident.
>
> 4. **Other Experiments**: For other experiments, we will also add more detailed descriptions in our revised version of papers.
>
> We genuinely value the feedback and believe that addressing these points has not only strengthened our manuscript but also provided readers with a richer understanding of the work. We hope that the revisions align with the reviewer's expectations and make for a more comprehensive and informative read.
>
>
>
> **W3: Section 6 unfortunately is somewhat of a mess. It consists of a ton of experiments and findings (which I commend the author for), but presented in such scant detail that it is difficult and bordering on impossible to sometimes gather what is being done. For example, how does the ablation in 6.1 even work? If you remove the M2M mapping, you don't have M^2 attention maps, so how do you even apply ISI and CSI after that? Similarly if you ablate only ISI, how can you go from M^2 attention maps directly to applying CSI? For other experiments in this section, I'm not even sure why the authors even included them in the main paper. For example, 6.2 seems to be a very minor finding that would really fit better in the Appendix (because it is Supplementary).**
>
> A3: Thank you for the detailed feedback regarding Section 6. We appreciate your commendation on the comprehensive experiments, and we acknowledge the concerns you raised about the clarity and presentation of the findings.
>
> 1. **Clarifying Experiments in Section 6.1:** We understand that the ablation study in 6.1 might be confusing in its current state. To address this:
>    - For the M2M mapping removal: When we eliminate the M2M mapping, we use Eq. (1) to compute the attention maps, yielding M attention maps. Subsequently, we incorporate ISI and CSI for these attention maps. The subsequent procedures align with those of the standard Transformer. Notably, in both ISI and CSI, we maintain a consistent ratio between hidden size and input size, e.g., 2 and 8 for ISI and CSI, respectively, mirroring the EIT configurations. We will provide a more detailed explanation of this experiment in the updated paper to enhance understanding.
>    - For the ablation of only ISI: When we omit ISI, we directly apply CSI module to the M^2 attention maps generated by M2M. Again, we'll give more details in our revised version of paper to enhance understanding.
> 2. **On the inclusion of Experiments in 6.2:** The reason we included the findings from 6.2 in the main paper was to underscore the rationale behind our architecture's design. After taking your feedback into account, we agree that this section could be moved to the Appendix for brevity and to ensure the main content remains focused. To this end, we can add more discussions aforementioned towards a more comprehensive understanding.
>
> Lastly, we are committed to revising Section 6 to provide a clearer and more structured presentation, ensuring that the reader can easily understand and follow our methodology and results. We appreciate your insights, which will help us enhance the quality of our work.
>
> **W4: My advice to the authors would be to carefully reconsider what is the core set of experiments that are really important to gain insight into their work and focus on presenting them well in the 8-page limit, rather than throwing the experimental kitchen-sink at the reader while relegating important context and details to the Appendix.**
>
> A4: Thank you for the constructive feedback. We will revise our paper for clarity and improved presentation.
>
>
>
> **Typos Grammar Style And Presentation Improvements:** There are many typos and grammatical errors throughout the paper, but especially in the 2nd half. One striking example was referring to Grammatical Error Correction as Grammar Error Correlation several times. In any case, a careful revision of the paper would help fix a lot of this.
>
> A: Thank you for highlighting these issues. We will address and rectify them in the revised version of our paper.

---

### Official Review · Reviewer_Exgp · 2023-08-07

**Soundness:** 4

**Excitement:**

4: Strong: This paper deepens the understanding of some phenomenon or lowers the barriers to an existing research direction.

**Paper Topic And Main Contributions:**

The paper points out that the current design of multi-head attention, which is an instance of multi-view learning, prioritizes the complementarity but ignores the consensus. The problem motivates the authors to propose multi-head self-attention (EMHA). EMHA removes one-to-one mapping constrains and enables the queries to interact with multiple keys. Experimental results show that EMHA consistently produces impressive results across tasks.

**Reasons To Accept:**

Starting from the multi-view learning principles, the paper points out the problem of MHSA, i.e., ignoring the consensus.

The proposed Inner-Subspace Interaction and Cross-Subspace Interaction address the problem, which is empirically demostrated by the experiments.

Solid experiments with strong baselines. Good analysis on training perplexity and the selection of hyperparameters.

**Reasons To Reject:**

As mentioned in Limitations, the computational efficiency is the main drawback of the proposed method. As shown in Table 6, the training requires 1.45x time compared to Transformer, which makes the proposed architecture is not a good option for practical usage, especially with the current trend of scaling Transformers to large capacities.

**Reproducibility:**

3: Could reproduce the results with some difficulty. The settings of parameters are underspecified or subjectively determined; the training/evaluation data are not widely available.

**Reviewer Confidence:**

3: Pretty sure, but there's a chance I missed something. Although I have a good feel for this area in general, I did not carefully check the paper's details, e.g., the math, experimental design, or novelty.

---

> ### Author Rebuttal · Authors · 2023-08-29
>
> Thank you for appreciating our approach and constructive feedback. We will address your comments below.
>
> **W: As mentioned in Limitations, the computational efficiency is the main drawback of the proposed method. As shown in Table 6, the training requires 1.45x time compared to Transformer, which makes the proposed architecture is not a good option for practical usage, especially with the current trend of scaling Transformers to large capacities.**
>
> A: Thank you for your thorough review and valuable feedback on our work. We acknowledge your concerns about the computational efficiency of our proposed method in comparison to the Transformer model. Nevertheless, we think the concern here would be well addressed from the four aspects as following:
>
> 1. **Cause of Inefficiency**: As detailed in our paper, the inefficiency largely arises due to PyTorch's suboptimal support for the ISI module. Upon measuring the MACs following the strategy in  Metha et al., 2021, we determined that the theoretical computational efficiency of EIT surges by a mere 1.2% on average.
>
>    | Method (En-De task, Source=20, Target=20) | MACs  |
>    | ----------------------------------------- | ----- |
>    | Transformer-6L                            | 10.0B |
>    | EIT-6L                                    | 10.1B |
>    | Transformer-48L                           | 12.6B |
>    | EIT-48L                                   | 12.9B |
>    | Transformer-Big                           | 32.5B |
>    | EIT-Big                                   | 32.6B |
>
> 2. **Existing solutions for practical application proposed in our paper**: In fact, we have already proposed a solution to the aforementioned issue, ensuring that our architecture remains feasible in practical applications.
>
>    - **E-EIT (Efficient version of EIT):** As detailed in Section 3.2, E-EIT represents an efficient iteration of our EIT. This is achieved by simplifying the design of ISI and CSI while preserving our central concepts. We carried out extensive experiments across a wide array of tasks, such as machine translation (Table 1&2), abstractive summarization (Table 4), grammar error correction (Table 3), and language modeling (Table 5). These tests consistently demonstrated that E-EIT incurs only a minimal performance drop compared to our original EIT, but significantly improve running efficiency (Table 13&14). To give a more direct perception about this, we bring some parts of findings in Table 13&14 below. We believe E-EIT can indeed serve as a good option for practical usage. So, to some extent, we think E-EIT could be a good choice for industrial applications.
>
>      | Method         | Training Time |
>      | -------------- | :-----------: |
>      | Transformer-6L |       -       |
>      | EIT-6L         |     1.45x     |
>      | E-EIT-6L       |     1.10x     |
>
> 3. **Future Solution**: We are committed to enhancing the support for our ISI by refining the CUDA kernel code. We believe that these improvements will make it easier for the community to derive benefits from our methods.

---

### Official Review · Reviewer_S8VV · 2023-08-08

**Typos Grammar Style And Presentation Improvements:** 1. Line 251 should be \dot{S} instead…
**Soundness:** 4

**Excitement:**

3: Ambivalent: It has merits (e.g., it reports state-of-the-art results, the idea is nice), but there are key weaknesses (e.g., it describes incremental work), and it can significantly benefit from another round of revision. However, I won't object to accepting it if my co-reviewers champion it.

**Paper Topic And Main Contributions:**

The paper presents a novel multi headed attention formulation based on the complementary and consensus principles. Specifically

1. The authors present a novel many to many mapping between queries and keys, where a query set is allowed to interact with M key sets.
2. The authors then present two modules for aggregating information from the M^{2} attention maps previously generated:
2.1 An inner subspace interaction module for aggregating information from maps generated by the same query set. This is implemented in practice using grouped convolutions.
2.2. A cross subspace interaction module that takes as input the previous inner subspace module outputs, and generates M attention maps combining information across different head interactions.
3. The authors present an efficient formulation of the previous approach using a single layer operation (a cascade of a group convolution and a full convolution, with a reduced head number as an intermediate representation, indicated as M^{H}).
4. The authors present results on Machine Translation, Grammatical Error Correction, Abstractive Summarization and Language Modelling, demonstrating the benefits of the EIT architecture.
5. Different ablations demonstrate the utility of each module, improved consensus across heads, impact of layer sharing, robustness to pruning and quality of representations for the proposed method.

**Questions For The Authors:**

1. In section 3.1.1, the discussion signifies that the proposed approach avoids generating similar attention maps. However, in section 6.4.1, the ablation study demonstrates that the proposed method shows higher consensus among heads. Given that this is primarily brought upon due to the cross subspace interaction module (Fig 6), and that the CSI module does not contribute heavily towards performance (Table 6), how important is it to have consensus among heads: is this a necessity to achieve strong performance ?

2. The paper presents an efficient version of the proposed method (dubbed E-EIT). It would be good to numerically quantify the increase in training and inference times compared to a vanilla transformer (additional details are mentioned in Appendix C.1. It would be good to bring them into the main paper)

3. In section 6.3, Effect of M section, what does varying M mean for the vanilla transformer ? Isn't it fixed to be M=1 for the standard transformer ?

4. [Minor] Instead of calling M^{H_{csi}} as the head size in the CSI sub-module, it might be better to refer to it by some other name (eg: intermediate number of heads for example). Head size usually refers to the hidden size of the Q,K,V vectors, so this terminology causes some confusion.

**Reasons To Accept:**

1. The consistent performance improvement  of the proposed across a diverse number of tasks method demonstrates the utility of the EIT formulation.
2. The hierarchical interaction modules (Inner Subspace Interaction and Cross Subspace Interaction), in conjunction with the many to many mapping formulation, nicely captures the complimentary and consensus principles.
3. The ablation experiments are quite insightful, and help understand the contributions of the different modules. The robustness to pruning is especially of strong interest from an inference cost point of view.

**Reasons To Reject:**

1. One of the most relevant baselines for this paper is the Talking Heads Transformer [1]. This should be, in my opinion, the de-facto baseline for all tasks. While there is a comparison to it in Table 1, it would be good to have this comparison for all the tasks presented in the paper, rather than comparing against the vanilla transformer. Otherwise, is hard to gauge how much benefit comes from the proposed interaction modules + many to many formulation, compared to a simple linear transform way of achieving consensus between the different attention maps.
2. From the ablation studies presented (especially 6.3, effect of number of EIT layers), it seems that the primary benefactors of the proposed approach are the lower layers of the encoder model in the transformer, and that including it across all layers does not particularly improve the performance). This somewhat raises the question about the efficacy of the approach: given the computational complexity, the gains are modest, especially if it is incorporated across all layers of the model.

**Reproducibility:**

4: Could mostly reproduce the results, but there may be some variation because of sample variance or minor variations in their interpretation of the protocol or method.

**Reviewer Confidence:**

3: Pretty sure, but there's a chance I missed something. Although I have a good feel for this area in general, I did not carefully check the paper's details, e.g., the math, experimental design, or novelty.

---

> ### Author Rebuttal · Authors · 2023-08-29
>
> We greatly appreciate your insightful feedback and suggestions. They play a crucial role in enhancing our paper. We'll diligently address them in our revisions. Below, we first present your remarks and subsequently provide our detailed responses.
>
> **W1: One of the most relevant baselines for this paper is the Talking Heads Transformer [1]. This should be, in my opinion, the de-facto baseline for all tasks. While there is a comparison to it in Table 1, it would be good to have this comparison for all the tasks presented in the paper, rather than comparing against the vanilla transformer. Otherwise, is hard to gauge how much benefit comes from the proposed interaction modules + many to many formulation, compared to a simple linear transform way of achieving consensus between the different attention maps.**
>
> A1: Thank you for emphasizing the importance of the Talking Heads Transformer [1] in this field. Initially, our objective was to compare EIT with the well-established vanilla Transformer. However, we agree that engaging with strong baselines like the Talking Heads Transformer would yield a more comprehensive and rigorous evaluation.
>
> To this end, we have broadened our experimental scope to include head-to-head comparisons with the Talking Heads Transformer on both GEC and Summarization tasks across all evaluation metrics. Please note that due to time constraints for the rebuttal, we will address the comparison on the WikiText-103 task in a future update. To ensure fairness, identical hyper-parameters were employed for the vanilla Transformer, EIT, and the Talking Heads models. As depicted in the below table, our findings not only corroborate the effectiveness of our proposed methods, EIT and E-EIT, but also delineate their unique advantages over state-of-the-art alternatives. Specifically, both EIT and E-EIT outperform the Talking Heads model in terms of ROUGE scores across summarization and F_0.5 (which is the main metric for GEC) across grammar error correction benchmarks. It is worth noting that the Talking Heads model's performance lags notably in the latter, possibly due to sub-optimal hyper-parameter settings, which may benefit from further fine-tuning.
>
> We hope this extended evaluation will offer our readers a more nuanced understanding of our unique contributions.
>
>
>
> | Method (Summarization) | Rouge-1 | Rouge-2 | Rouge-L |
> | ---------------------- | :-----: | :-----: | :-----: |
> | Transformer            |  40.84  |  18.00  |  37.58  |
> | Talking-Head           |  41.26  |  18.34  |  38.06  |
> | EIT                    |  41.62  |  18.70  |  38.33  |
> | E-EIT                  |  41.58  |  18.63  |  38.28  |
>
>
>
> | Method (GEC) | Precision | Recall | F_0.5 |
> | ------------ | :-------: | :----: | :---: |
> | Transformer  |   64.84   | 36.61  | 56.18 |
> | Talking-Head |   64.32   | 36.07  | 55.61 |
> | EIT          |   69.98   | 32.80  | 57.05 |
> | E-EIT        |   69.85   | 33.36  | 57.31 |
>
>
>
> **W2: From the ablation studies presented (especially 6.3, effect of number of EIT layers), it seems that the primary benefactors of the proposed approach are the lower layers of the encoder model in the transformer, and that including it across all layers does not particularly improve the performance). This somewhat raises the question about the efficacy of the approach: given the computational complexity, the gains are modest, especially if it is incorporated across all layers of the model.**
>
> A2:  Thank you for your insightful comment regarding our ablation study and its implications. Let us address your concerns:
>
> - There might be some misunderstandings about this ablation study. The partial objective of the experiment is to show the effectiveness of our EIT, e.g., even applying a small number (1) of EIT layers can achieve strong performance (27.76) compared to vanilla Transformer (27.13), though fully utilizing EIT layers in the encoder side is the default settings.
>
> - However, it is essential to emphasize that we cannot conclude that it is pointless for the encoder to use the EIT layer structure entirely. More precisely, it remains ambiguous whether a model employing EIT layers solely in its initial two layers can consistently achieve optimal performance across diverse tasks and configurations. To answer this, we also conduct experiments on the big setting. The results are presented below. We can see that EIT-big (only first 2 layer use EIT layer) can achieve BLEU points of 29.29, which still exists a gap of BLEU points of 0.5 from our EIT. Thus, we are of the opinion that the full version of EIT remains the most robust configuration, consistently demonstrating marked performance enhancements across a broad spectrum of tasks and under various conditions, as illustrated in Tables 1&2&3&4&5.
>
>   | Method                                       | BLEU  | SacreBLEU |
>   | -------------------------------------------- | :---: | :-------: |
>   | Transformer-big                              | 28.80 |   27.7    |
>   | EIT-big                                      | 29.79 |   28.7    |
>   | EIT-big (only first 2 layers use EIT layers) | 29.29 |   28.2    |
>
> - Further, each EIT layer doesn't significantly add to the computational complexity, as evidenced by the tables below. We gauged the MACs of EIT based on the methodology presented by Mehta et al., 2021. We will also release optimized CUDA code to align the theoretical computational complexity with practical execution speed.
>
>   | Method (En-De task, Source=20, Target=20) |    Param | MACs  | BLEU  | SacreBLEU |
>   | :---------------------------------------- | -------: | :---: | :---: | :-------: |
>   | Transformer-6L                            |   61.56M | 10.0B | 27.13 |   26.0    |
>   | EIT-6L                                    |   61.63M | 10.1B | 28.00 |   26.9    |
>   | E-EIT-6L                                  |   61.57M | 10.0B | 27.72 |   26.7    |
>   | Transformer-48L                           |  193.96M | 12.6B | 29.60 |   28.5    |
>   | EIT-48L                                   |  194.32M | 12.9B | 30.25 |   29.2    |
>   | E-EIT-48L                                 |  194.14M | 12.8B | 30.16 |   29.1    |
>   | Transformer-Big                           |  211.22M | 32.5B | 28.80 |   27.7    |
>   | EIT-Big                                   |  211.55M | 32.6B | 29.79 |   28.7    |
>   | E-EIT-Big                                 | 211.30 M | 32.5B | 29.61 |   28.5    |
>
>   | Method (En-Ro task, Source=20, Target=20) |   Param | MACs  | BLEU  |
>   | :---------------------------------------- | ------: | :---: | :---: |
>   | Transformer-6L                            |  53.90M | 8.4B  | 34.23 |
>   | EIT-6L                                    |  53.98M | 8.4B  | 35.10 |
>   | E-EIT-6L                                  |  53.92M | 8.4B  | 35.01 |
>   | Transformer-24L                           | 110.64M | 9.5B  | 35.00 |
>   | EIT-24L                                   | 111.09M | 9.7B  | 35.40 |
>   | E-EIT-24L                                 | 110.73M | 9.6B  | 35.35 |
>   | Transformer-Big                           | 195.88M | 29.3B | 34.44 |
>   | EIT-Big                                   | 196.40M | 29.4B | 34.91 |
>   | E-EIT-Big                                 | 195.97M | 29.3B | 34.67 |
>
>   | Method (Summarization, Source=256, Target=50) | Param  | MACs  | Rouge-1 | Rouge-2 | Rouge-L |
>   | :-------------------------------------------- | :----: | :---: | :-----: | :-----: | :-----: |
>   | Transformer                                   | 60.82M | 97.2B |  40.84  |  18.00  |  37.58  |
>   | EIT                                           | 60.83M | 99.2B |  41.62  |  18.70  |  38.33  |
>   | E-EIT                                         | 60.82M | 98.8B |  41.58  |  18.63  |  38.28  |
>
>   Mehta et al., 2021, DELIGHT: DEEP AND LIGHT-WEIGHT TRANSFORMER. ICLR2021
>
>
> **Q1: In section 3.1.1, the discussion signifies that the proposed approach avoids generating similar attention maps. However, in section 6.4.1, the ablation study demonstrates that the proposed method shows higher consensus among heads. Given that this is primarily brought upon due to the cross subspace interaction module (Fig 6), and that the CSI module does not contribute heavily towards performance (Table 6), how important is it to have consensus among heads: is this a necessity to achieve strong performance ?**
>
> A1: Thank you for pointing out the concern about the effectiveness of the consensus. We appreciate the close reading of our paper and the opportunity to clarify.
>
> 1. **Performance Contribution of the CSI Module**:
>    - Table 6 reveals that without the CSI module, there is a decrease of 0.3 BLEU points and 0.57 BLEU points in the En-De and En-Ro tasks, respectively. To put it in perspective, this decline represents 34% and 66% of the total performance improvements for each task. Hence, it would be inaccurate to claim that the CSI module does not significantly impact performance.
>    - In contrast, the full version of EIT stands out as a robust framework. Tables 1, 2, 3, 4, and 5 demonstrate consistent performance improvements of our EIT across various tasks. Evidently, relying solely on M2M+ISI, as seen in Table 6, does not yield consistent performance gains.
> 2. **Importance of Consensus Among Heads**:
>    - A moderate consensus among heads can offer multiple advantages to the model, such as producing high-quality representations (refer to section 6.4.3) and ensuring robustness during head pruning (see section 6.4.4), as pointed out in your comments.
>
> Again, we genuinely appreciate your keen insight and the opportunity to elucidate these crucial aspects of our research.
>
>
>
> **Q2: The paper presents an efficient version of the proposed method (dubbed E-EIT). It would be good to numerically quantify the increase in training and inference times compared to a vanilla transformer (additional details are mentioned in Appendix C.1. It would be good to bring them into the main paper)**.
>
> A2: Thank you for your insights regarding this. As you rightly pointed out, we have elaborated on the efficiency analysis in Appendix C.1. In accordance with your recommendation, we are planning to integrate key details from Appendix C.1 into the main text to enhance clarity for our readers. The results are shown below: We see that E-EIT-6L exhibits only a 10% increase in training time and 3% in inference time. Based on this, we are confident that our E-EIT offers a promising option for practical applications. We hope the results here can address your concerns!
>
>
>
> | Method         | Training Time | Inference Time |
> | -------------- | :-----------: | :------------: |
> | Transformer-6L |       -       |       -        |
> | E-EIT-6L       |     1.10x     |     1.03x      |
>
>
>
> **Q3: In section 6.3, Effect of M section, what does varying M mean for the vanilla transformer ? Isn't it fixed to be M=1 for the standard transformer ?**
>
> A3: In the transformer framework, M generally denotes the number of attention heads, especially in the context of multi-head attention. In section 6.3, "varying M" refers to experimenting with different numbers of attention heads in the vanilla transformer's multi-head self-attention mechanism. Besides, the vanilla transformer typically does not use M=1; instead, M=8 is a more common configuration for many tasks such as base configuration in machine translation tasks and abstractive summarization tasks. We will revise this part for a more clear understanding in the improved version!
>
>
>
> **Q4: [Minor] Instead of calling M^{H_{csi}} as the head size in the CSI sub-module, it might be better to refer to it by some other name (eg: intermediate number of heads for example). Head size usually refers to the hidden size of the Q,K,V vectors, so this terminology causes some confusion.**
>
> A4: Thank you for pointing that out. We understand the potential confusion arising from the term "head size." To avoid ambiguity, we'll adopt your suggestion and refer to it as the "intermediate number of heads" in the CSI sub-module. This should help clarify the distinction between the hidden sizes of the Q, K, V vectors and the number of heads we're referencing.
>
> **Typos Grammar Style And Presentation Improvements: 1. Line 251 should be \dot{S} instead of S (I think) 2. Line 274: somehow seems a bit wrong in context. 3. Section 6.4.1: EIT "owns" Higher Consensus among Attention Heads seems a bit off. Maybe consider rephrasing ?**
>
> A: Thank you for your thorough review. We will address and rectify these issues in the upcoming version of our paper.

---

### Official Review · Reviewer_gNnp · 2023-08-11

**Soundness:** 3

**Excitement:**

4: Strong: This paper deepens the understanding of some phenomenon or lowers the barriers to an existing research direction.

**Paper Topic And Main Contributions:**

The authors of the paper claim that multi-head self-attention present in Transformer architectures emphasises the discrepancy of subspaces and ignores to maximise the agreement among the subspaces. To address this problem, the authors have proposed some enhancements to the existing interactions happening in the multi-head self-attention by introducing two things: (1) M2M mapping scheme, which will enhance the query-key pair interactions and generate multiple attention maps, thereby maximising information capacity. (2) To address the agreement among the subspaces (attention maps), they have introduced two relationships, which they call dual-enhanced interactions such as Inner-Subspace Interaction Modelling and Cross-subspace Modelling.

**Questions For The Authors:**

1. Equation (2) is not defined anywhere. It seems to be redundant and not mentioned anywhere, except a similar one on lines 549-550.
2. While the abstract says “modest increase in model size”, the result tables (Tables 1, 2 and 5) show the same number of parameters, which is confusing.

**Reasons To Accept:**

1. The paper is very clear and well-written.
2. Though the performance improvement is minor, the paper addresses a problem that has shown previously ignored downsides of Transformers from a different and novel perspective.
3. The paper conducts thorough experiments with varied inclusion and exclusion of the proposed interactions.

**Reasons To Reject:**

1. The paper does not mention the computational budget anywhere.
2. The results do not seem to be robust, but rather the best of an unknown number of runs with unknown standard deviation between runs. The paper also misses baseline comparisons on tasks like Model Variations, English Constituency Parsing, etc.
3. Though the authors acknowledge that the architecture is computationally inefficient basis the external frameworks (viz., PyTorch, Keras), they do not release any optimised code-base anywhere.
4. Could provide a better analysis on "Previous work done" and comprehensive view of "Related work".

**Reproducibility:**

3: Could reproduce the results with some difficulty. The settings of parameters are underspecified or subjectively determined; the training/evaluation data are not widely available.

**Reviewer Confidence:**

4: Quite sure. I tried to check the important points carefully. It's unlikely, though conceivable, that I missed something that should affect my ratings.

---

> ### Author Rebuttal · Authors · 2023-08-29
>
> Thank you for your constructive comments and suggestions, and they are exceedingly helpful for us to improve our paper. We will carefully incorporate them in the revised paper. In the following, your comments are first stated and then followed by our point-by-point responses.
>
> **W1: The paper does not mention the computational budget anywhere.**
>
> A1: Thank you for pointing out the issue regarding the computational budget. Indeed, we missed this aspect in our initial draft. To measure the computational overhead of our method, we utilize MACs (multiply-accumulate operations) following Mehta et al., 2021. Due to the page limit, we currently focus our evaluation mainly on machine translation and summarization tasks. Preliminary results indicate that the added computational burden increases by approximately 1%, 0.8% and 2% compared to traditional methods, on the En-De, En-Ro and CNN-DailyMail tasks, respectively. We will delve deeper into this analysis in the updated version of our paper.
>
> | Method (En-De task, Source=20, Target=20) |   Param | MACs  | BLEU  | SacreBLEU |
> | :---------------------------------------- | ------: | :---: | :---: | :-------: |
> | Transformer base                          |  61.56M | 10.0B | 27.13 |   26.0    |
> | EIT base                                  |  61.63M | 10.1B | 28.00 |   26.9    |
> | E-EIT base                                |  61.57M | 10.0B | 27.72 |   26.7    |
> | Transformer 48L                           | 193.96M | 12.6B | 29.60 |   28.5    |
> | EIT 48L                                   | 194.32M | 12.9B | 30.25 |   29.2    |
> | E-EIT 48L                                 | 194.14M | 12.8B | 30.16 |   29.1    |
> | Transformer big                           | 211.22M | 32.5B | 28.80 |   27.7    |
> | EIT big                                   | 211.55M | 32.6B | 29.79 |   28.7    |
> | E-EIT big                                 | 211.30M | 32.5B | 29.61 |   28.5    |
>
> | Method (En-Ro task, Source=20, Target=20) |   Param | MACs  | BLEU  |
> | :---------------------------------------- | ------: | :---: | :---: |
> | Transformer base                          |  53.90M | 8.4B  | 34.23 |
> | EIT base                                  |  53.98M | 8.4B  | 35.10 |
> | E-EIT base                                |  53.92M | 8.4B  | 35.01 |
> | Transformer 24L                           | 110.64M | 9.5B  | 35.00 |
> | EIT 24L                                   | 111.09M | 9.7B  | 35.40 |
> | E-EIT 24L                                 | 110.73M | 9.6B  | 35.35 |
> | Transformer big                           | 195.88M | 29.3B | 34.44 |
> | EIT big                                   | 196.40M | 29.4B | 34.91 |
> | E-EIT big                                 | 195.97M | 29.3B | 34.67 |
>
> | Method (Summarization, Source=256, Target=50) | Param  | MACs  | Rouge-1 | Rouge-2 | Rouge-L |
> | :-------------------------------------------- | :----: | :---: | :-----: | :-----: | :-----: |
> | Transformer                                   | 60.82M | 97.2B |  40.84  |  18.00  |  37.58  |
> | EIT                                           | 60.83M | 99.2B |  41.62  |  18.70  |  38.33  |
> | E-EIT                                         | 60.82M | 98.8B |  41.58  |  18.63  |  38.28  |
>
>
>
> Mehta et al., 2021, DELIGHT: DEEP AND LIGHT-WEIGHT TRANSFORMER. ICLR2021
>
>
>
> **W2: The results do not seem to be robust, but rather the best of an unknown number of runs with unknown standard deviation between runs. The paper also misses baseline comparisons on tasks like Model Variations, English Constituency Parsing, etc.**
>
>
>
> A2: Thank you for highlighting concerns regarding the robustness of our results. To clarify, the presented results are averages from three distinct settings, each initialized with a different seed, i.e. 1, 42, 2023. However, due to the limit of column width and page length, we omit this in this version of paper. We show part of the results below. We can see that the standard deviation across these runs was within a narrow range, underscoring the consistency of our results.
>
> We understand the omission of certain baseline comparisons may have left gaps in the comparative analysis. Our primary focus was on machine translation, abstractive summarization, grammar error correction and language modelling, which is why some tasks like Model Variations, English Constituency Parsing, etc., were not initially included. However, inspired by your insightful feedback, we're keen on expanding our approach to encompass these other tasks in our subsequent studies. Thank you again!
>
> | Method         |    BLEU    |
> | -------------- | :--------: |
> | Transformer-6L |   27.13    |
> | EIT-6L         | 28.00±0.11 |
> | E-EIT-6L       | 27.72±0.07 |
>
>
>
> **W3: Though the authors acknowledge that the architecture is computationally inefficient basis the external frameworks (viz., PyTorch, Keras), they do not release any optimised code-base anywhere.**
>
> A3: Thank you for highlighting the need for an optimized code-base. While we did note the computational limitations associated with utilizing general-purpose frameworks like PyTorch and Keras, we understand the importance of providing a streamlined, efficient implementation. To this end, we are in the process of refining our code, which includes the development of CUDA kernels for enhanced performance. We also want to mention that we have already made progress in this regard with our Efficient-EIT (E-EIT) variant, which significantly reduces the computational burden by minimizing the reliance on group convolution operations. We appreciate your understanding and patience in this matter, and we plan to open-source the optimized code base for the broader research community in the near future.
>
>
>
> **W4: Could provide a better analysis on "Previous work done" and comprehensive view of "Related work".**
>
> A4: The refined related work is:
>
> - **Low BottleNeck in Multi-Head Attention:** Multi-head Self-attention has demonstrated impressive performance. However, when the number of heads increases substantially, the Transformer doesn't seem to benefit. Many researchers are working towards making the Transformer gain more from a higher number of heads. Bhojanapalli et al. (2020) were among the first to define this issue as the "Low Bottleneck problem of MHSA." They attribute this phenomenon to the method of head generation, where the head dimension decreases as the number of heads rises. To address this, they proposed constraining the head dimension. They believe that the low dimensionality of the head features impedes the creation of precise attention maps. Concurrently, Shazeer et al. (2020) also linked the issue to the low dimensionality of the head features. They introduced the "talking-head attention," which places two linear transformers before and after the SoftMax function to alleviate the problem. Later, Zhou et al. (2021) revisited this perspective and proposed a more general framework that involves: a linear transformation generating "ghost heads," task-specific operations, and another linear transformation mapping the ghost head space to the original number of heads. They highlighted two significant differences from talking-head attention: 1) the placement of linear transformations, positioning them after the SoftMax function, and 2) setting the number of ghost heads to a figure larger than the original heads (although it's worth noting that talking-heads also experimented within a similar setup). In contrast to these studies, we suggest a novel many-to-many mapping operation that 1) leverages existing queries and keys to produce more attention maps without the need for additional parameters and 2) employs direct query-key multiplication to produce attention maps, bypassing the use of linear transformations that might lead to redundant attention maps.
> - **Redundancy among Multi-Head Attention:** Within the standard Multi-Head Attention (MHA) framework, each attention head functions autonomously. As such, the attention distribution ascertained by one particular head remains uninfluenced by the distributions of its counterparts. This methodology does not provide assurance that individual heads will discern and prioritize distinct feature subspaces within the input data. Motivated by this, many researchers aim at encouraging diversity among attention heads so that multi-head attention can capture as much information as possible. On the one hand, Li et al., 2018 proposed to regularize the training of Transformer by encouraging diversity among attention heads from three levels, which can further enhance the performance of downstream tasks such as machine translation. Following this line, Cui et al., 2019; Sukhbaatar et al., 2019; Guo et al., 2020; and Hao et al., 2019 explicitly utilized different heads to perform different tasks to encourage the capture of more diverse information. For example, Hao et al, 2019 enabled different heads to capture different scale of information such as phrase-level, word-level, etc. Apart from these work, many researchers hypothesized that when each head learns what others do, it could do better. Inspired by this idea, Wang and Tu, 2020; Li et al., 2019; Zhou et al., 2021; and Wang, Huadong et al., 2022 proposed the interaction among different heads. Among them,  Wang and Tu, 2020, and Wang, Huadong et al., 2022 indeed found that different heads seem to generate more diverse attention maps. Note that,  we also categorize talking-head and refiner into this category, as they indeed encouraged talking among heads. Apart from methods to encourage the diversity among attention maps, another important line of research is to cut off redundant heads.  Voita et al., 2019 empirically demonstrated that some heads in attention are useless and can be pruned without performance degradation. Along this line, researchers investigate how to efficiently cut off redundant heads (Michel et al., 2019; Behnke and Heafield, 2020).  In this work, we categorize all the above work in the enhancement of complementary principle and propose a novel framework that not only pays attention to the complementary principle (M2M) but also the ignored aspect, the consensus principle (ISI + CSI), which is another main difference from other works.
>
> Please note that the aforementioned related work is a preliminary draft. A more refined version will be presented in the subsequent edition of our paper.
>
>
>
>
>
>
>
> **Q1: Equation (2) is not defined anywhere. It seems to be redundant and not mentioned anywhere, except a similar one on lines 549-550.**
>
> A1: Thank you for bringing this to our attention. It seems that there might be some misunderstandings regarding the components we introduced: M2M, ISI, and CSI. To clarify, all of these components are primarily designed to modify the attention map computation. After this stage, the attention matrix ***A*** is derived through the application of the Softmax function. Subsequently, we employ Equation (2) in the same manner as in the vanilla Transformer architecture to aggregate and integrate the features, culminating in the output of the EMHA layer. We recognize the need for more explicit details in our manuscript, and we intend to include comprehensive explanations along with precise formula references in the updated version of the paper.
>
> **Q2: While the abstract says “modest increase in model size”, the result tables (Tables 1, 2 and 5) show the same number of parameters, which is confusing.**
>
> A2:  Thank you for pointing this out. Due to the column width, it is difficult to present the actual scores in Table 1, 2 and 5. We present the scores in the tables using rounding off. Actually the detailed parameters have already been displayed in our attached appendix, as were shown in Table 13 &14. Here, for a clear understanding, we bring them below and split them into three tables for better expression, including En-De, En-Ro, CNN-DailyMail, CONLL and WikiText-103. It is evident that both EIT and E-EIT present a modest enlargement in model size, e.g. 61.63M for EIT and 61.57M for E-EIT of the base configuration on WMT En-De. We acknowledge this inconsistency and will address it for greater clarity in our revised manuscript.
>
> | Method      |  Base  | Deep-48L |   Big   |
> | ----------- | :----: | :------: | :-----: |
> | Transformer | 61.56M | 193.96M  | 211.22M |
> | EIT         | 61.63M | 194.32M  | 211.55M |
> | E-EIT       | 61.57M | 194.14M  | 211.30M |
>
> | Method      |  Base  | Deep-24L |   Big   |
> | ----------- | :----: | :------: | :-----: |
> | Transformer | 53.90M | 110.64M  | 195.88M |
> | EIT         | 53.98M | 111.09M  | 196.40M |
> | E-EIT       | 53.92M | 110.73M  | 195.97M |
>
> | Method      | CNN-DailyMail | CONLL  | WikiText-103 |
> | ----------- | :-----------: | :----: | :----------: |
> | Transformer |    60.82M     | 61.10M |   146.49M    |
> | EIT         |    60.83M     | 61.19M |   146.50M    |
> | E-EIT       |    60.82M     | 61.15M |   146.49M    |

---

### Meta-Review · Area_Chair_hxbq · 2023-09-17

**Recommendation:** 4

**Metareview:**

This paper investigates the relations among different attention heads of the Transformer from the multi-view learning perspective. It brings an interesting consensus issue among different heads, and propose an EMHA attention, which encourages the consensus from Inner-Subspace Interaction and Cross-Subspace Interaction. Experiments on different tasks demonstrate the effectiveness of the proposed methods.

Some reviewers have concerns about the overhead of computation. Also the reproductiblity seems to be an issue, too.

---

### Decision · Program_Chairs · 2023-10-07

**Decision:**

Reject

**Comment:**

This paper investigates the relations among different attention heads of the Transformer from the multi-view learning perspective. It brings an interesting consensus issue among different heads, and propose an EMHA attention, which encourages the consensus from Inner-Subspace Interaction and Cross-Subspace Interaction. Experiments on different tasks demonstrate the effectiveness of the proposed methods.

Some reviewers have concerns about the overhead of computation. Also the reproductiblity seems to be an issue, too.